# Independent SE(3)-Equivariant Models for End-to-End Rigid Protein Docking

**Octavian-Eugen Ganea**[†*]
MIT

**Xinyuan Huang**[§*]
ETH Zurich

**Charlotte Bunne**
ETH Zurich

**Yatao Bian**[†]
Tencent AI Lab

**Regina Barzilay**
MIT

**Tommi Jaakkola**
MIT

**Andreas Krause**
ETH Zurich

## Abstract

Protein complex formation is a central problem in biology, being involved in most of the cell's processes, and essential for applications, e.g. drug design or protein engineering. We tackle *rigid body protein-protein docking*, i.e., computationally predicting the 3D structure of a protein-protein complex from the individual unbound structures, assuming no conformational change within the proteins happens during binding. We design a novel pairwise-independent SE(3)-equivariant graph matching network to predict the rotation and translation to place one of the proteins at the right docked position relative to the second protein. We mathematically guarantee a basic principle: the predicted complex is always identical regardless of the initial locations and orientations of the two structures. Our model, named EQUIDOCK, approximates the binding pockets and predicts the docking poses using keypoint matching and alignment, achieved through optimal transport and a differentiable Kabsch algorithm. Empirically, we achieve significant running time improvements and often outperform existing docking software despite not relying on heavy candidate sampling, structure refinement, or templates. [1]

## 1 Introduction

In a recent breakthrough, ALPHAFOLD 2 (Jumper et al., 2021; Senior et al., 2020) provides a solution to a grand challenge in biology—inferring a protein's three-dimensional structure from its amino acid sequence (Baek et al., 2021), following the dogma *sequence determines structure*.

Besides their complex three-dimensional nature, proteins dynamically alter their function and structure in response to cellular signals, changes in the environment, or upon *molecular docking*. In par-

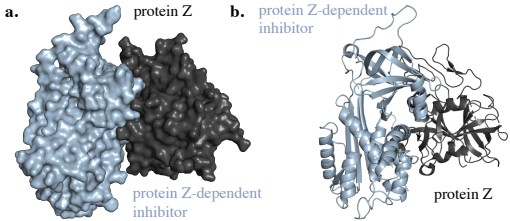

Figure 1: **Different views of the 3D structure of a protein complex. a.** Surface and **b.** cartoon view of protein Z and its inhibitor.

ticular, protein interactions are involved in various biological processes including signal transduction, protein synthesis, DNA replication and repair. Molecular docking is key to understanding protein interactions' mechanisms and effects, and, subsequently, to developing therapeutic interventions.

We here address the problem of *rigid body protein-protein docking* which refers to computationally predicting the 3D structure of a protein-protein complex given the 3D structures of the two proteins in unbound state. *Rigid body* means no deformations occur within any protein during binding, which is a realistic assumption in many biological settings.

Popular docking software (Chen et al., 2003; Venkatraman et al., 2009; De Vries et al., 2010; Torchala et al., 2013; Schindler et al., 2017; Sunny and Jayaraj, 2021) are typically computationally expensive,

---

[†]Correspondence to: Octavian Ganea (oct@mit.edu) and Yatao Bian (yatao.bian@gmail.com).
[*]Equal contribution.
[§]Work done during an internship at Tencent AI Lab.
[1]Our code is publicly available: https://github.com/octavian-ganea/equidock_public.

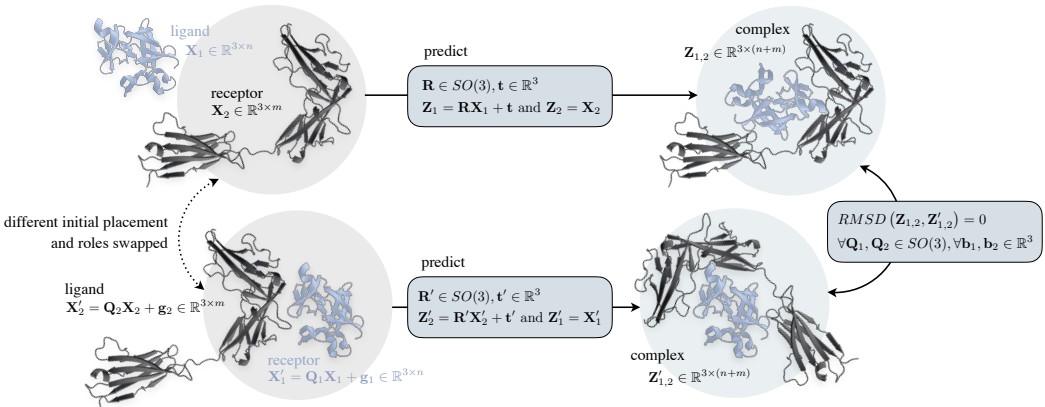

Figure 2: **Same output guarantee of EQUIDOCK.** We predict a rigid transformation to place the ligand in the binding location w.r.t the receptor. We mathematically guarantee to output the same complex structure — up to an SE(3) transformation — independently of the initial unbound positions, rotations, or roles of both constituents. (RMSD = Root-mean-square deviation of atomic positions)

taking between minutes and hours to solve a single example pair, while not being guaranteed to find accurate complex structures. These methods largely follow the steps: i.) randomly sample a large number (e.g., millions) of candidate initial complex structures, ii.) employ a scoring function to rank the candidates, iii.) adjust and refine the top complex structures based on an energy model (e.g., force field). We here take a first step towards tackling these issues by using deep learning models for direct prediction of protein complex structures.

**Contributions.** We design EQUIDOCK, a fast, end-to-end method for rigid body docking that directly predicts the SE(3) transformation to place one of the proteins (ligand) at the right location and orientation with respect to the second protein (receptor). Our method is based on the principle that the exact same complex structure should be predicted irrespectively of the initial 3D placements and roles of both constituents (see Fig. 2). We achieve this desideratum by incorporating the inductive biases of pairwise SE(3)–equivariance and commutativity, and deriving novel theoretical results for necessary and sufficient model constraints (see Section 3). Next, we create EQUIDOCK to satisfy these properties by design, being a combination of: i) a novel type of pairwise independent SE(3)-equivariant graph matching networks, ii) an attention-based keypoint selection algorithm that discovers representative points and aligns them with the binding pocket residues using optimal transport, and iii) a differentiable superimposition model to recover the optimal global rigid transformation. Unlike prior work, our method does not use heavy candidate sampling or ranking, templates, task-specific geometric or chemical hand-crafted features, or pre-computed meshes. This enables us to achieve plausible structures with a speed-up of 80-500x compared to popular docking software, offering a promising competitive alternative to current solutions for this problem.

## 2 RELATED WORK

**Geometric Deep Learning.** Graph Neural Networks (GNNs) are becoming the de facto choice for learning with graph data (Bruna et al., 2013; Defferrard et al., 2016; Kipf and Welling, 2016; Gilmer et al., 2017; Xu et al., 2018; Li et al., 2019). Motivated by symmetries naturally occurring in different data types, architectures are tailored to explicitly incorporate such properties (Cohen and Welling, 2016a;b; Thomas et al., 2018; Fuchs et al., 2020; Finzi et al., 2020; Eismann et al., 2020; Satorras et al., 2021). GNNs are validated in a variety of tasks such as particle system dynamics or conformation-based energy estimation (Weiler and Cesa, 2019; Rezende et al., 2019).

**Euclidean Neural Networks (E(3)-NNs).** However, plain GNNs and other deep learning methods do not understand data naturally lying in the 3D Euclidean space. For example, how should the output deterministically change with the input, e.g. when it is rotated ? The recent Euclidean neural networks address this problem, being designed from geometric first-principles. They make use of SE(3)- equivariant and invariant neural layers, thus avoiding expensive data augmentation strategies. Such constrained models ease optimization and have shown important improvements in biology or chemistry – e.g. for molecular structures (Fuchs et al., 2020; Hutchinson et al., 2020; Wu et al., 2021;

Jumper et al., 2021; Ganea et al., 2021) and different types of 3D point clouds (Thomas et al., 2018). Different from prior work, we here derive constraints for pairs of 3D objects via *pairwise independent SE(3)-equivariances*, and design a principled approach for modeling rigid body docking.

**Protein Folding.** Deep neural networks have been used to predict inter-residue contacts, distance and/or orientations (Adhikari and Cheng, 2018; Yang et al., 2020; Senior et al., 2020; Ju et al., 2021), that are subsequently transformed into additional constraints or differentiable energy terms for protein structure optimization. ALPHAFOLD 2 (Jumper et al., 2021) and Rosetta Fold (Baek et al., 2021) are state-of-the-art approaches, and directly predict protein structures from co-evolution information embedded in homologous sequences, using geometric deep learning and E(3)-NNs.

**Protein-Protein Docking and Interaction.** Experimentally determining structures of protein complexes is often expensive and time-consuming, rendering a premium on computational methods (Vakser, 2014). Protein docking methods (Chen et al., 2003; Venkatraman et al., 2009; De Vries et al., 2010; Biesiada et al., 2011; Torchala et al., 2013; Schindler et al., 2017; Weng et al., 2019; Sunny and Jayaraj, 2021; Christoffer et al., 2021; Yan et al., 2020) typically run several steps: first, they sample thousands or millions of complex candidates; second, they use a scoring function for ranking (Moal et al., 2013; Basu and Wallner, 2016; Launay et al., 2020; Eismann et al., 2020); finally, top-ranked candidates undergo a structure refinement process using energy or geometric models (Verburgt and Kihara, 2021). Relevant to protein-protein interaction (PPI) is the task of protein interface prediction where GNNs have showed promise (Fout et al., 2017; Townshend et al., 2019; Liu et al., 2020; Xie and Xu, 2021; Dai and Bailey-Kellogg, 2021). Recently, ALPHAFOLD 2 and ROSETTAFOLD have been utilized as subroutines to improve PPIs from different aspects (Humphreys et al., 2021; Pei et al., 2021; Jovine), e.g., combining physics-based docking method CLUSPRO (Kozakov et al., 2017; Ghani et al., 2021), or using extended multiple-sequence alignments to predict the structure of heterodimeric protein complexes from the sequence information (Bryant et al., 2021). Concurrently to our work, Evans et al. (2021) extend ALPHAFOLD 2 to multiple chains during both training and inference.

**Drug-Target Interaction (DTI).** DTI aims to compute drug-target binding poses and affinity, playing an essential role in understanding drugs' mechanism of action. Prior methods (Wallach et al., 2015; Li et al., 2021) predict binding affinity from protein-ligand co-crystal structures, but such data is expensive to obtain experimentally. These models are typically based on heavy candidate sampling and ranking (Trott and Olson, 2010; Koes et al., 2013; McNutt et al., 2021; Bao et al., 2021), being tailored for small drug-like ligands and often assuming known binding pocket. Thus, they are not immediately applicable to our use case. In contrast, our rigid docking approach is generic and could be extended to accelerate DTI research as part of future work.

## 3 MATHEMATICAL CONSTRAINTS FOR RIGID BODY DOCKING

We start by introducing the rigid body docking problem and derive the geometric constraints for enforcing same output complex prediction regardless of the initial unbound positions or roles (Fig. 2).

**Rigid Protein-Protein Docking – Problem Setup.** We are given as input a pair of proteins forming a complex. They are (arbitrarily) denoted as the ligand and receptor, consisting of $n$ and $m$ residues, respectively. These proteins are represented in their bound (docked) state as 3D point clouds $\mathbf{X}_1^* \in \mathbb{R}^{3 \times n}, \mathbf{X}_2^* \in \mathbb{R}^{3 \times m}$, where each residue's location is given by the coordinates of its corresponding $\alpha$-carbon atom. In the unbound state, the docked ligand is randomly rotated and translated in space, resulting in a modified point cloud $\mathbf{X}_1 \in \mathbb{R}^{3 \times n}$. For simplicity and w.l.o.g., the receptor remains in its bound location $\mathbf{X}_2 = \mathbf{X}_2^*$.

*The task is to predict a rotation $\mathbf{R} \in SO(3)$ and a translation $\mathbf{t} \in \mathbb{R}^3$ such that $\mathbf{R}\mathbf{X}_1 + \mathbf{t} = \mathbf{X}_1^*$, using as input the proteins and their unbound positions $\mathbf{X}_1$ and $\mathbf{X}_2$.*

Here, $\mathbf{R} = \mathbf{R}(\mathbf{X}_1|\mathbf{X}_2)$ and $\mathbf{t} = \mathbf{t}(\mathbf{X}_1|\mathbf{X}_2)$ are functions of the two proteins, where we omit residue identity or other protein information in this notation, for brevity.

Note that we assume rigid backbone and side-chains for both proteins. We therefore do not tackle the more challenging problem of *flexible docking*, but our approach offers an important step towards it.

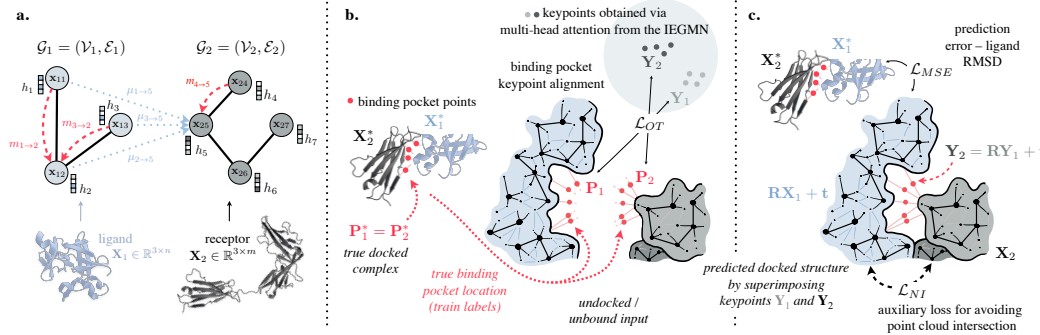

Figure 3: **Details on EQUIDOCK's Architecture and Losses. a.** The message passing operations in IEGMN guarantee pairwise independent SE(3)-equivariance as in Eq. (4), **b.** We predict keypoints for each protein that are aligned with the binding pocket location using an additional optimal transport (OT) loss, **c.** After predicting the docked position, we compute an MSE loss on the ligand, as well as a loss to discourage body intersections.

We desire that the predicted complex structure is independent of the initial locations and orientations of the two proteins, as well as of their roles – see Fig. 2. Formally, we wish to guarantee that:

$$
\begin{aligned}
(\mathbf{R}(\mathbf{Z}_1|\mathbf{Z}_2)\mathbf{Z}_1 + \mathbf{t}(\mathbf{Z}_1|\mathbf{Z}_2)) \oplus \mathbf{Z}_2 &\equiv (\mathbf{R}(\mathbf{X}_1|\mathbf{X}_2)\mathbf{X}_1 + \mathbf{t}(\mathbf{X}_1|\mathbf{X}_2)) \oplus \mathbf{X}_2, \quad \text{(SE(3)-invariance)} \\
(\mathbf{R}(\mathbf{X}_1|\mathbf{X}_2)\mathbf{X}_1 + \mathbf{t}(\mathbf{X}_1|\mathbf{X}_2)) \oplus \mathbf{X}_2 &\equiv \mathbf{X}_1 \oplus (\mathbf{R}(\mathbf{X}_2|\mathbf{X}_1)\mathbf{X}_2 + \mathbf{t}(\mathbf{X}_2|\mathbf{X}_1)), \quad \text{(commutativity)} \\
\forall \mathbf{Q}_1, \mathbf{Q}_2 \in SO(3), \forall \mathbf{g}_1, \mathbf{g}_2 \in \mathbb{R}^3, &\forall \mathbf{X}_1 \in \mathbb{R}^{3 \times n}, \mathbf{X}_2 \in \mathbb{R}^{3 \times m}, \text{ and } \mathbf{Z}_l = \mathbf{Q}_l \mathbf{X}_l + \mathbf{g}_l, l \in \{1, 2\}.
\end{aligned}
\tag{1}
$$

for any rotations $\mathbf{Q}_1, \mathbf{Q}_2$ and translations $\mathbf{g}_1, \mathbf{g}_2$, where $\oplus$ is concatenation along columns, and $\equiv$ denotes identity after superimposition, i.e. zero Root-Mean-Square Deviation (RMSD) between the two 3D point sets after applying the Kabsch algorithm (Kabsch, 1976). An immediate question arises:

*How do the constraints in Eq. (1) translate into constraints for $\mathbf{R}(\cdot|\cdot)$ and $\mathbf{t}(\cdot|\cdot)$ ?*

The rotation $\mathbf{R}$ and translation $\mathbf{t}$ change in a systematic way when we apply $SE(3)$ transformations or swap proteins' roles. These properties restrict our class of functions as derived below.

**SE(3)-equivariance Constraints.** If we apply any distinct $SE(3)$ transformations on the unbound ligand $\mathbf{X}_1$ and receptor $\mathbf{X}_2$, i.e. we dock $\mathbf{Q}_1\mathbf{X}_1 + \mathbf{g}_1$ onto $\mathbf{Q}_2\mathbf{X}_2 + \mathbf{g}_2$, then the rotation matrix $\mathbf{R}(\mathbf{Q}_1\mathbf{X}_1 + \mathbf{g}_1|\mathbf{Q}_2\mathbf{X}_2 + \mathbf{g}_2)$ and translation vector $\mathbf{t}(\mathbf{Q}_1\mathbf{X}_1 + \mathbf{g}_1|\mathbf{Q}_2\mathbf{X}_2 + \mathbf{g}_2)$ can be derived from the original $\mathbf{R}(\mathbf{X}_1|\mathbf{X}_2)$ and $\mathbf{t}(\mathbf{X}_1|\mathbf{X}_2)$ assuming that we always do rotations first. In this case, $\mathbf{R}(\mathbf{Q}_1\mathbf{X}_1 + \mathbf{g}_1|\mathbf{Q}_2\mathbf{X}_2 + \mathbf{g}_2)$ can be decomposed into three rotations: i.) apply $\mathbf{Q}_1^\top$ to undo the rotation $\mathbf{Q}_1$ applied on $\mathbf{X}_1$, ii.) apply $\mathbf{R}(\mathbf{X}_1|\mathbf{X}_2)$, iii.) apply $\mathbf{Q}_2$ to rotate the docked ligand together with the receptor. This gives $\mathbf{R}(\mathbf{Q}_1\mathbf{X}_1 + \mathbf{g}_1|\mathbf{Q}_2\mathbf{X}_2 + \mathbf{g}_2) = \mathbf{Q}_2\mathbf{R}(\mathbf{X}_1|\mathbf{X}_2)\mathbf{Q}_1^\top$, which in turn constraints the translation vector. We provide a formal statement and prove it in Appendix B.1:

**Proposition 1.** *For any $\mathbf{Q}_1, \mathbf{Q}_2 \in SO(3), \mathbf{g}_1, \mathbf{g}_2 \in \mathbb{R}^3$, SE(3)-invariance of the predicted docked complex defined by Eq. (1) is guaranteed iff*

$$
\begin{aligned}
\mathbf{R}(\mathbf{Q}_1\mathbf{X}_1 + \mathbf{g}_1|\mathbf{Q}_2\mathbf{X}_2 + \mathbf{g}_2) &= \mathbf{Q}_2\mathbf{R}(\mathbf{X}_1|\mathbf{X}_2)\mathbf{Q}_1^\top \\
\mathbf{t}(\mathbf{Q}_1\mathbf{X}_1 + \mathbf{g}_1|\mathbf{Q}_2\mathbf{X}_2 + \mathbf{g}_2) &= \mathbf{Q}_2\mathbf{t}(\mathbf{X}_1|\mathbf{X}_2) - \mathbf{Q}_2\mathbf{R}(\mathbf{X}_1|\mathbf{X}_2)\mathbf{Q}_1^\top\mathbf{g}_1 + \mathbf{g}_2.
\end{aligned}
\tag{2}
$$

As a direct consequence of this proposition, we have the following statement.

**Proposition 2.** *Any model satisfying Proposition 1 guarantees invariance of the predicted complex w.r.t. any SE(3) transformation on $\mathbf{X}_1$, and equivariance w.r.t. any SE(3) transformation on $\mathbf{X}_2$:*

$$
\begin{aligned}
\mathbf{R}(\mathbf{Z}_1|\mathbf{X}_2)\mathbf{Z}_1 + \mathbf{t}(\mathbf{Z}_1|\mathbf{X}_2) &= \mathbf{R}(\mathbf{X}_1|\mathbf{X}_2)\mathbf{X}_1 + \mathbf{t}(\mathbf{X}_1|\mathbf{X}_2), \quad \text{where } \mathbf{Z}_1 = \mathbf{Q}_1\mathbf{X}_1 + \mathbf{g}_1 \\
\mathbf{R}(\mathbf{X}_1|\mathbf{Z}_2)\mathbf{X}_1 + \mathbf{t}(\mathbf{X}_1|\mathbf{Z}_2) &= \mathbf{Q}_2\left[\mathbf{R}(\mathbf{X}_1|\mathbf{X}_2)\mathbf{X}_1 + \mathbf{t}(\mathbf{X}_1|\mathbf{X}_2)\right] + \mathbf{g}_2, \quad \text{where } \mathbf{Z}_2 = \mathbf{Q}_2\mathbf{X}_2 + \mathbf{g}_2 \\
\forall \mathbf{Q}_1, \mathbf{Q}_2 \in SO(3), &\forall \mathbf{g}_1, \mathbf{g}_2 \in \mathbb{R}^3, \forall \mathbf{X}_1 \in \mathbb{R}^{3 \times n}, \forall \mathbf{X}_2 \in \mathbb{R}^{3 \times m}.
\end{aligned}
$$

**Commutativity.** Instead of docking $\mathbf{X}_1$ with respect to $\mathbf{X}_2$, we can also dock $\mathbf{X}_2$ with respect to $\mathbf{X}_1$. In this case, we require the final complex structures to be identical after superimposition, i.e., zero RMSD. This property is named *commutativity* and it is satisfied as follows (proof in Appendix B.2).

**Proposition 3.** *Commutativity as defined by Eq.* (1) *is guaranteed iff*

$$\mathbf{R}(\mathbf{X}_2|\mathbf{X}_1) = \mathbf{R}^\top(\mathbf{X}_1|\mathbf{X}_2); \quad \mathbf{t}(\mathbf{X}_2|\mathbf{X}_1) = -\mathbf{R}^\top(\mathbf{X}_1|\mathbf{X}_2)\mathbf{t}(\mathbf{X}_1|\mathbf{X}_2), \tag{3}$$

**Point Permutation Invariance.** We also enforce residue permutation invariance. Formally, both $\mathbf{R}(\mathbf{X}_1|\mathbf{X}_2)$ and $\mathbf{t}(\mathbf{X}_1|\mathbf{X}_2)$ should not depend on the order or columns of $\mathbf{X}_1$ and, resp., of $\mathbf{X}_2$.

## 4 EQUIDOCK MODEL

**Protein Representation.** A protein is a sequence of amino acid residues that folds in a 3D structure. Each residue has a general structure with a side-chain specifying its type, allowing us to define a *local frame* and derive SE(3)-invariant features for any pair of residues —see Appendix A.

We represent a protein as a graph $\mathcal{G} = (\mathcal{V}, \mathcal{E})$, similar to Fout et al. (2017); Townshend et al. (2019); Liu et al. (2020). Each node $i \in \mathcal{V}$ represents one residue and has 3D coordinates $\mathbf{x}_i \in \mathbb{R}^3$ corresponding to the $\alpha$-carbon atom's location. Edges are given by a k-nearest-neighbor (k-NN) graph using Euclidean distance of the original 3D node coordinates.

**Overview of Our Approach.** Our model is depicted in Fig. 3. We first build k-NN protein graphs $\mathcal{G}_1 = (\mathcal{V}_1, \mathcal{E}_1)$ and $\mathcal{G}_2 = (\mathcal{V}_2, \mathcal{E}_2)$. We then design SE(3)-invariant node features $\mathbf{F}_1 \in \mathbb{R}^{d \times n}, \mathbf{F}_2 \in \mathbb{R}^{d \times m}$ and edge features $\{\mathbf{f}_{j \to i} : \forall (i, j) \in \mathcal{E}_1 \cup \mathcal{E}_2\}$ (see Appendix A).

Next, we apply several layers consisting of functions $\Phi$ that jointly transform node coordinates and features. Crucially, we guarantee, by design, *pairwise independent SE(3)-equivariance* for coordinate embeddings, and invariance for feature embeddings. This double constraint is formally defined:

Given $\mathbf{Z}_1, \mathbf{H}_1, \mathbf{Z}_2, \mathbf{H}_2 = \Phi(\mathbf{X}_1, \mathbf{F}_1, \mathbf{X}_2, \mathbf{F}_2)$

we have $\mathbf{Q}_1\mathbf{Z}_1 + \mathbf{g}_1, \mathbf{H}_1, \mathbf{Q}_2\mathbf{Z}_2 + \mathbf{g}_2, \mathbf{H}_2 = \Phi(\mathbf{Q}_1\mathbf{X}_1 + \mathbf{g}_1, \mathbf{F}_1, \mathbf{Q}_2\mathbf{X}_2 + \mathbf{g}_2, \mathbf{F}_2),$ (4)

$\forall \mathbf{Q}_1, \mathbf{Q}_2 \in SO(3), \forall \mathbf{g}_1, \mathbf{g}_2 \in \mathbb{R}^3.$

We implement $\Phi$ as a novel type of message-passing neural network (MPNN). We then use the output node coordinate and feature embeddings to compute $\mathbf{R}(\mathbf{X}_1|\mathbf{X}_2)$ and $\mathbf{t}(\mathbf{X}_1|\mathbf{X}_2)$. These functions depend on pairwise interactions between the two proteins modeled as cross-messages, but also incorporate the 3D structure in a pairwise-independent SE(3)-equivariant way to satisfy Eq. (1), Proposition 1 and Proposition 3. We discover keypoints from each protein based on a neural attention mechanism and softly guide them to represent the respective binding pocket locations via an optimal transport based auxiliary loss. Finally, we obtain the SE(3) transformation by superimposing the two keypoint sets via a differentiable version of the Kabsch algorithm. An additional soft-constraint discourages point cloud intersections. We now detail each of these model components.

**Independent E(3)-Equivariant Graph Matching Networks (IEGMNs).** Our architecture for $\Phi$ satisfying Eq. (4) is called *Independent E(3)-Equivariant Graph Matching Network* (IEGMN) – see Fig. 3. It extends both Graph Matching Networks (GMN) (Li et al., 2019) and E(3)-Equivariant Graph Neural Networks (E(3)-GNN) (Satorras et al., 2021). IEGMNs perform node coordinate and feature embedding updates for an input pair of graphs $\mathcal{G}_1 = (\mathcal{V}_1, \mathcal{E}_1), \mathcal{G}_2 = (\mathcal{V}_2, \mathcal{E}_2)$, and use inter- and intra-node messages, as well as E(3)-equivariant coordinate updates. The $l$-th layer of IEGMNs transforms node latent/feature embeddings $\{\mathbf{h}_i^{(l)}\}_{i \in \mathcal{V}_1 \cup \mathcal{V}_2}$ and node coordinate embeddings $\{\mathbf{x}_i^{(l)}\}_{i \in \mathcal{V}_1 \cup \mathcal{V}_2}$ as

$$\mathbf{m}_{j \to i} = \varphi^e(\mathbf{h}_i^{(l)}, \mathbf{h}_j^{(l)}, \exp(-\|\mathbf{x}_i^{(l)} - \mathbf{x}_j^{(l)}\|^2/\sigma), \mathbf{f}_{j \to i}), \forall e_{j \to i} \in \mathcal{E}_1 \cup \mathcal{E}_2 \tag{5}$$

$$\boldsymbol{\mu}_{j \to i} = a_{j \to i} \mathbf{W} \mathbf{h}_j^{(l)}, \forall i \in \mathcal{V}_1, j \in \mathcal{V}_2 \text{ or } i \in \mathcal{V}_2, j \in \mathcal{V}_1 \tag{6}$$

$$\mathbf{m}_i = \frac{1}{|\mathcal{N}(i)|} \sum_{j \in \mathcal{N}(i)} \mathbf{m}_{j \to i}, \forall i \in \mathcal{V}_1 \cup \mathcal{V}_2 \tag{7}$$

$$\boldsymbol{\mu}_i = \sum_{j \in \mathcal{V}_2} \boldsymbol{\mu}_{j \to i}, \forall i \in \mathcal{V}_1, \quad \text{and} \quad \boldsymbol{\mu}_i = \sum_{j \in \mathcal{V}_1} \boldsymbol{\mu}_{j \to i}, \forall i \in \mathcal{V}_2 \tag{8}$$

$$\mathbf{x}_i^{(l+1)} = \eta \mathbf{x}_i^{(0)} + (1 - \eta)\mathbf{x}_i^{(l)} + \sum_{j \in \mathcal{N}(i)} (\mathbf{x}_i^{(l)} - \mathbf{x}_j^{(l)})\varphi^x(\mathbf{m}_{j \to i}), \forall i \in \mathcal{V}_1 \cup \mathcal{V}_2 \tag{9}$$

$$\mathbf{h}_i^{(l+1)} = (1 - \beta) \cdot \mathbf{h}_i^{(l)} + \beta \cdot \varphi^h(\mathbf{h}_i^{(l)}, \mathbf{m}_i, \boldsymbol{\mu}_i, \mathbf{f}_i), \forall i \in \mathcal{V}_1 \cup \mathcal{V}_2, \tag{10}$$

where $\mathcal{N}(i)$ are the neighbors of node $i$; $\varphi^x$ is a real-valued (scalar) parametric function; $\mathbf{W}$ is a learnable matrix; $\varphi^h, \varphi^e$ are parametric functions (MLPs) outputting a vector $\mathbb{R}^d$; $\mathbf{f}_{j \to i}$ and $\mathbf{f}_i$ are the original edge and node features (extracted SE(3)-invariantly from the residues). $a_{j \to i}$ is an attention based coefficient with trainable shallow neural networks $\psi^q$ and $\psi^k$:

$$a_{j \to i} = \frac{\exp(\langle \psi^q(\mathbf{h}_i^{(l)}), \psi^k(\mathbf{h}_j^{(l)}) \rangle)}{\sum_{j'} \exp(\langle \psi^q(\mathbf{h}_i^{(l)}), \psi^k(\mathbf{h}_{j'}^{(l)}) \rangle)}, \tag{11}$$

Note that all parameters of $\mathbf{W}, \varphi^x, \varphi^h, \varphi^e, \psi^q, \psi^k$ can be shared or different for different IEGMN layers . The output of several IEGMN layers is then denoted as:

$$\mathbf{Z}_1 \in \mathbb{R}^{3 \times n}, \mathbf{H}_1 \in \mathbb{R}^{d \times n}, \mathbf{Z}_2 \in \mathbb{R}^{3 \times m}, \mathbf{H}_2 \in \mathbb{R}^{d \times m} = IEGMN(\mathbf{X}_1, \mathbf{F}_1, \mathbf{X}_2, \mathbf{F}_2). \tag{12}$$

It is then straightforward to prove the following (see Appendix B.3):

**Proposition 4.** *IEGMNs satisfy the pairwise independent SE(3)-equivariance property in Eq. (4).*

**Keypoints for Differentiable Protein Superimposition.** Next, we use multi-head attention to obtain $K$ points for each protein, $\mathbf{Y}_1, \mathbf{Y}_2 \in \mathbb{R}^{3 \times K}$, which we name **keypoints**. We train them to become representative points for the binding pocket of the respective protein pair (softly-enforced by an additional loss described later). If this would holds perfectly, then the superimposition of $\mathbf{Y}_1$ and $\mathbf{Y}_2$ would give the corresponding ground truth superimposition of $\mathbf{X}_1$ and $\mathbf{X}_2$. Our model is :

$$\mathbf{y}_{1k} := \sum_{i=1}^n \alpha_i^k \mathbf{z}_{1i}; \quad \mathbf{y}_{2k} := \sum_{j=1}^m \beta_j^k \mathbf{z}_{2j},$$

where $\mathbf{z}_{1i}$ denotes the i-th column of matrix $\mathbf{Z}_1$, and $\alpha_i^k = softmax_i(\frac{1}{\sqrt{d}} \mathbf{h}_{1i}^\top \mathbf{W}_k' \mu(\varphi(\mathbf{H}_2)))$ are attention scores (similarly defined for $\beta_j^k$), with $\mathbf{W}_k' \in \mathbb{R}^{d \times d}$ a parametric matrix (different for each attention head), $\varphi$ a linear layer plus a LeakyReLU non-linearity, and $\mu(\cdot)$ is the mean vector.

**Differentiable Kabsch Model.** We design the rotation and translation that docks protein 1 into protein 2 to be the same transformation used to superimpose $\mathbf{Y}_1$ and $\mathbf{Y}_2$ — see Fig. 3. For this, we compute a differentiable version of the Kabsch algorithm (Kabsch, 1976) as follows. Let $\mathbf{A} = \overline{\mathbf{Y}}_2 \overline{\mathbf{Y}}_1^\top \in \mathbb{R}^{3 \times 3}$ computed using zero-mean keypoints. The singular value decomposition (SVD) is $\mathbf{A} = \mathbf{U}_2 \mathbf{S} \mathbf{U}_1^\top$, where $\mathbf{U}_2, \mathbf{U}_1 \in O(3)$. Finally, we define the differentiable functions

$$\mathbf{R}(\mathbf{X}_1 | \mathbf{X}_2; \theta) = \mathbf{U}_2 \begin{pmatrix} 1 & 0 & 0 \\ 0 & 1 & 0 \\ 0 & 0 & d \end{pmatrix} \mathbf{U}_1^\top, \quad \text{where } d = \text{sign}(\det(\mathbf{U}_2 \mathbf{U}_1^\top))$$

$$\mathbf{t}(\mathbf{X}_1 | \mathbf{X}_2; \theta) = \mu(\mathbf{Y}_2) - \mathbf{R}(\mathbf{X}_1 | \mathbf{X}_2; \theta) \mu(\mathbf{Y}_1), \tag{13}$$

where $\mu(\cdot)$ is the mean vector of a point cloud. It is straightforward to prove that this model satisfies all the equivariance properties in Eqs. (1) to (3). From a practical perspective, the gradient and backpropagation through the SVD operation was analyzed by (Ionescu et al., 2015; Papadopoulo and Lourakis, 2000) and implemented in the automatic differentiation frameworks such as PyTorch.

**MSE Loss.** During training, we randomly decide which protein is the receptor (say protein 2), keep it in the docked position (i.e., $\mathbf{X}_2 = \mathbf{X}_2^*$), predict the SE(3) transformation using Eq. (13) and use it to compute the final position of the ligand as $\tilde{\mathbf{X}}_1 = \mathbf{R}(\mathbf{X}_1 | \mathbf{X}_2)\mathbf{X}_1 + \mathbf{t}(\mathbf{X}_1 | \mathbf{X}_2)$. The mean squared error (MSE) loss is then $\mathcal{L}_{\text{MSE}} = \frac{1}{n} \sum_{i=1}^n \|\mathbf{x}_i^* - \tilde{\mathbf{x}}_i\|^2$.

**Optimal Transport and Binding Pocket Keypoint Alignment.** As stated before, we desire that $\mathbf{Y}_1$ and $\mathbf{Y}_2$ are representative points for the binding pocket location of the respective protein pair. However, this needs to be encouraged explicitly, which we achieve using an additional loss.

We first define the binding pocket point sets, inspiring from previous PPI work (Section 2). Given the residues' $\alpha$-carbon locations of the bound (docked) structures, $\mathbf{X}_1^*$ and $\mathbf{X}_2^*$, we select all pairs of residues at less than $\tau$ Euclidean distance ($\tau = 8\text{Å}$ in our experiments). We assume these are all interacting residues. Denote these pairs as $\{(\mathbf{x}_{1s}^*, \mathbf{x}_{2s}^*), s \in 1, \dots, S\}$, where $S$ is variable across data pairs. We compute midpoints of these segments, denoted as $\mathbf{P}_1^*, \mathbf{P}_2^* \in \mathbb{R}^{3 \times S}$, where

$\mathbf{p}_{1s}^* = \mathbf{p}_{2s}^* = 0.5 \cdot (\mathbf{x}_{1s}^* + \mathbf{x}_{2s}^*)$. We view $\mathbf{P}_1^*$ and $\mathbf{P}_2^*$ as binding pocket points. In the unbound state, these sets are randomly moved in space together with the respective protein residue coordinates $\mathbf{X}_1$ and $\mathbf{X}_2$. We denote them as $\mathbf{P}_1, \mathbf{P}_2 \in \mathbb{R}^{3 \times S}$. For clarity, if $\mathbf{X}_1 = \mathbf{Q}\mathbf{X}_1^* + \mathbf{g}$, then $\mathbf{P}_1 = \mathbf{Q}\mathbf{P}_1^* + \mathbf{g}$.

We desire that $\mathbf{Y}_1$ is a representative set for the 3D set $\mathbf{P}_1$ (and, similarly, $\mathbf{Y}_2$ for $\mathbf{P}_2$). However, while at training time we know that every point $\mathbf{p}_{1s}$ corresponds to the point $\mathbf{p}_{2s}$ (and, similarly, $\mathbf{y}_{1k}$ aligns with $\mathbf{y}_{2k}$, by assumption), we unfortunately do not know the actual alignment between points in $\mathbf{Y}_l$ and $\mathbf{P}_l$, for every $l \in \{1, 2\}$. This can be recovered using an additional optimal transport loss:

$$\mathcal{L}_{\mathrm{OT}} = \min_{\mathbf{T} \in \mathcal{U}(S,K)} \langle \mathbf{T}, \mathbf{C} \rangle, \quad \text{where } \mathbf{C}_{s,k} = \|\mathbf{y}_{1k} - \mathbf{p}_{1s}\|^2 + \|\mathbf{y}_{2k} - \mathbf{p}_{2s}\|^2, \tag{14}$$

where $\mathcal{U}(S, K)$ is the set of $S \times K$ transport plans with uniform marginals. The optimal transport plan is computed using an Earth Mover's Distance and the POT library (Flamary et al., 2021), while being kept fixed during back-propagation and optimization when only the cost matrix is trained.

Note that our approach assumes that $\mathbf{y}_{1k}$ corresponds to $\mathbf{y}_{2k}$, for every $k \in \{1, \ldots, K\}$. Intuitively, each attention head $k$ will identify a specific geometric/chemical local surface feature of protein 1 by $\mathbf{y}_{1k}$, and match its complementary feature of protein 2 by $\mathbf{y}_{2k}$.

**Avoiding Point Cloud Intersection.** In practice, our model does not enforce a useful inductive bias, namely that proteins forming complexes are never "intersecting" with each other. To address this issue, we first state a notion of the "interior" of a protein point cloud. Following previous work (Sverrisson et al., 2021; Venkatraman et al., 2009), we define the surface of a protein point cloud $\mathbf{X} \in \mathbb{R}^{3 \times n}$ as $\{\mathbf{x} \in \mathbb{R}^3 : G(\mathbf{x}) = \gamma\}$, where $G(\mathbf{x}) = -\sigma \ln(\sum_{i=1}^{n} \exp(-\|\mathbf{x} - \mathbf{x}_i\|^2/\sigma))$. The parameters $\sigma$ and $\gamma$ are chosen such that there exist no "holes" inside a protein (we found $\gamma = 10, \sigma = 25$ to work well, see Appendix E). As a consequence, the interior of the protein is given by $\{\mathbf{x} \in \mathbb{R}^3 : G(\mathbf{x}) < \gamma\}$. Then, the condition for non-intersecting ligand and receptor can be written as $G_1(\mathbf{x}_{2j}) > \gamma, \forall j \in 1, \ldots, m$ and $G_2(\mathbf{x}_{1i}) > \gamma, \forall i \in 1, \ldots, n$. As a loss function, this becomes

$$\mathcal{L}_{\mathrm{NI}} = \frac{1}{n} \sum_{i=1}^{n} \max(0, \gamma - G_2(\mathbf{x}_{1i})) + \frac{1}{m} \sum_{j=1}^{m} \max(0, \gamma - G_1(\mathbf{x}_{2j})). \tag{15}$$

**Surface Aware Node Features.** Surface contact modeling is important for protein docking. We here design a novel surface feature type that differentiates residues closer to the surface of the protein from those in the interior. Similar to Sverrisson et al. (2021), we prioritize efficiency and avoid pre-computing meshes, but show that our new feature is a good proxy for residue's depth (i.e. distance to the protein surface). Intuitively, residues in the core of the protein are locally surrounded *in all directions* by other residues. This is not true for residues on the surface, e.g., neighbors are in a half-space if the surface is locally flat. Building on this intuition, for each node (residue) $i$ in the $k$-NN protein graph, we compute the norm of the weighted average of its neighbor forces, which can be interpreted as the normalized gradient of the $G(\mathbf{x})$ surface function. This SE(3)-invariant feature is

$$\rho_i(\mathbf{x}_i; \lambda) = \frac{\|\sum_{i' \in \mathcal{N}_i} w_{i,i',\lambda}(\mathbf{x}_i - \mathbf{x}_{i'})\|}{\sum_{i' \in \mathcal{N}_i} w_{i,i',\lambda}\|\mathbf{x}_i - \mathbf{x}_{i'}\|}, \quad \text{where } w_{i,i',\lambda} = \frac{\exp(-\|\mathbf{x}_i - \mathbf{x}_{i'}\|^2/\lambda)}{\sum_{j \in \mathcal{N}_i} \exp(-\|\mathbf{x}_i - \mathbf{x}_j\|^2/\lambda)}. \tag{16}$$

Intuitively, as depicted in Fig. 8, residues in the interior of the protein have values close to 0 since they are surrounded by vectors from all directions that cancel out, while residues near the surface have neighbors only in a narrower cone, with aperture depending on the local curvature of the surface. We show in Appendix C that this feature correlates well with more expensive residue depth estimation methods, e.g. based on MSMS, thus offering a computationally appealing alternative. We also compute an estimation of this feature for large dense point clouds based on the local surface angle.

## 5 EXPERIMENTS

**Datasets.** We leverage the following datasets: Docking Benchmark 5.5 (DB5.5) (Vreven et al., 2015) and Database of Interacting Protein Structures (DIPS) (Townshend et al., 2019). DB5.5 is a gold standard dataset in terms of data quality, but contains only 253 structures. DIPS is a larger protein complex structures dataset mined from the Protein Data Bank (Berman et al., 2000) and tailored for rigid body docking. Datasets information is given in Appendix D. We filter DIPS to only

Table 1: **Complex Prediction Results.** As in the main text, the proprietary baselines might internally use parts of the test sets (e.g. to extract templates or features), thus their numbers might be optimistic.

| | **DIPS Test Set** | | | | | | **DB5.5 Test Set** | | | | | |
| | Complex RMSD | | | Interface RMSD | | | Complex RMSD | | | Interface RMSD | | |
| **Methods** | Median | Mean | Std | Median | Mean | Std | Median | Mean | Std | Median | Mean | Std |
| ATTRACT | 17.17 | 14.93 | 10.39 | 12.41 | 14.02 | 11.81 | 9.55 | 10.09 | 9.88 | 7.48 | 10.69 | 10.90 |
| HDOCK | 6.23 | 10.77 | 11.39 | 3.90 | 8.88 | 10.95 | 0.30 | 5.34 | 12.04 | 0.24 | 4.76 | 10.83 |
| CLUSPRO | 15.76 | 14.47 | 10.24 | 12.54 | 13.62 | 11.11 | 3.38 | 8.25 | 7.92 | 2.31 | 8.71 | 9.89 |
| PATCHDOCK | 15.24 | 13.58 | 10.30 | 11.44 | 12.15 | 10.50 | 18.26 | 18.00 | 10.12 | 18.88 | 18.75 | 10.06 |
| **EQUIDOCK** | 13.29 | 14.52 | 7.13 | 10.18 | 11.92 | 7.01 | 14.13 | 14.72 | 5.31 | 11.97 | 13.23 | 4.93 |

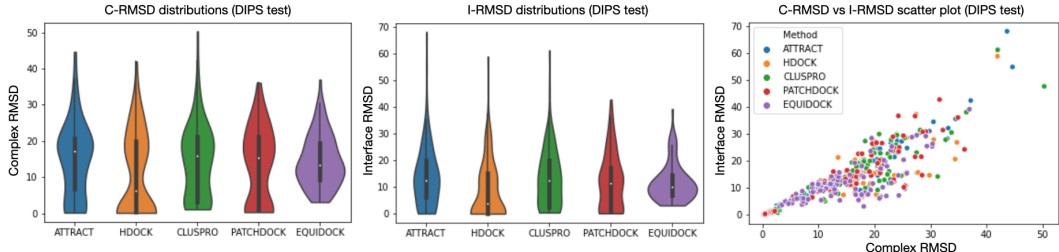

Figure 4: **a.** Complex-RMSD distributions (DIPS test set); **b.** Interface-RMSD distributions (DIPS test set); **c.** scatter plot for C-RMSD vs I-RMSD (DIPS test set).

keep proteins with at most 10K atoms. Datasets are then randomly partitioned in train/val/test splits of sizes 203/25/25 (DB5.5) and 39,937/974/965 (DIPS). For DIPS, the split is based on protein family to separate similar proteins. For the final evaluation in Table 1, we use the full DB5.5 test set, and randomly sample 100 pairs from different protein families from the DIPS test set.

**Baselines.** We compare our EQUIDOCK method with popular state-of-the-art docking software [2] CLUS-PRO (PIPER) (Desta et al., 2020; Kozakov et al., 2017), ATTRACT (Schindler et al., 2017; de Vries et al., 2015), PATCHDOCK (Mashiach et al., 2010; Schneidman-Duhovny et al., 2005), and HDOCK (Yan et al., 2020; 2017b;a; Huang and Zou, 2014; 2008). These baselines provide user-friendly local packages suitable for automatic experiments or webservers for manual submissions.

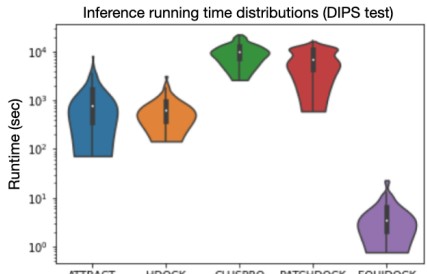

Figure 5: Inference running time distributions (log10 scale).

**Evaluation Metrics.** To measure prediction's quality, we report Complex Root Mean Square Deviation (CRMSD) and Interface Root Mean Square Deviation (IRMSD), defined below. Given the ground truth and predicted complex structures, $\mathbf{Z}^* \in \mathbb{R}^{3 \times (n+m)}$ and $\mathbf{Z} \in \mathbb{R}^{3 \times (n+m)}$, we first superimpose them using the Kabsch algorithm (Kabsch, 1976), and then compute C-RMSD $= \sqrt{\frac{1}{n+m} \|\mathbf{Z}^* - \mathbf{Z}\|_F^2}$. We compute I-RMSD similarly, but using only the coordinates of the interface residues with distance less than 8Å to the other protein's residues. For a fair comparison among baselines, we use only the $\alpha$-carbon coordinates to compute both metrics.

**Training Details.** We train our models on the train part of DIPS first, using Adam (Kingma and Ba, 2014) with learning rate 2e-4 and early stopping with patience of 30 epochs. We update the best validation model only when it achieves a score of less than 98% of the previous best validation score, where the score is the median of Ligand RMSD on the full DIPS validation set. The best DIPS validated model is then tested on the DIPS test set. For DB5.5, we fine tune the DIPS pre-trained

---
[2]ClusPro: https://cluspro.bu.edu/, Attract: www.attract.ph.tum.de/services/ATTRACT/ATTRACT.vdi.gz, PatchDock: https://bioinfo3d.cs.tau.ac.il/PatchDock/, HDOCK: http://huanglab.phys.hust.edu.cn/software/HDOCK/

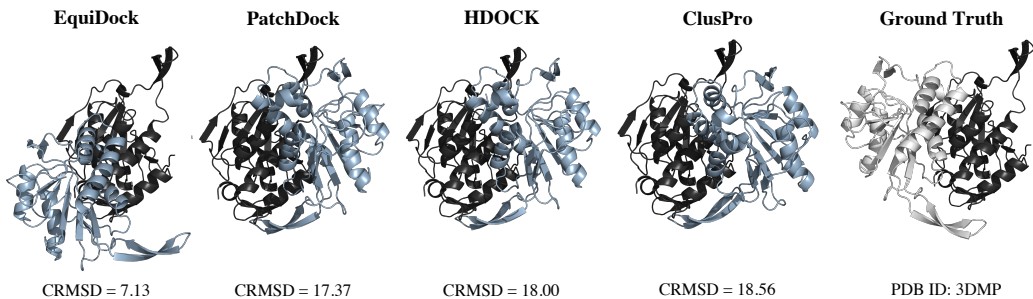

Figure 6: Visualization of a protein complex successfully predicted by EQUIDOCK. Note that all other methods find the binding interface on the wrong side of the black protein.

model on the DB5.5 training set using learning rate 1e-4 and early stopping with 150 epochs patience. The best DB5.5 validated model is finally tested on DB5.5 test set. During training, we randomly assign the roles of ligand and receptor. Also, during both training and testing, we randomly rotate and translate the ligand in space (even though our model is invariant to this operation) for all baselines.

**Complex Prediction Results.** Results are shown in Table 1, Fig. 4 and Appendix E. We note that our method is competitive and often outperforms the baselines. However, we do not use heavy candidate sampling and re-ranking, we do not rely on task-specific hand-crafted features, and we currently do not perform structure fine-tuning, aiming to predict the SE(3) ligand transformation in a direct shot. Moreover, we note that some of the baselines might have used part of our test set in validating their models, for example to learn surface templates, thus, their reported scores might be optimistic. Notably, HDOCK score function was validated on DB4 which overlaps with DB5.5. A more appropriate comparison would require us to re-build these baselines without information from our test sets, a task that is currently not possible without open-source implementations.

**Computational Efficiency.** We show inference times in Fig. 5 and Table 4. Note that EQUIDOCK is between 80-500 times faster than the baselines. This is especially important for intensive screening applications that aim to scan over vast search spaces, e.g. for drug discovery. In addition, it is also relevant for de novo design of binding proteins (e.g. antibodies (Jin et al., 2021)) or for use cases when protein docking models are just a component of significantly larger end-to-end architectures targeting more involved biological scenarios, e.g., representing a drug's mechanism of action or modeling cellular processes with a single model as opposed to a multi-pipeline architecture.

**Visualization.** We show in Fig. 6 a successful example of a test DIPS protein pair for which our model significantly outperforms all baselines.

## 6  CONCLUSION

We have presented an extremely fast, end-to-end rigid protein docking approach that does not rely on candidate sampling, templates, task-specific features or pre-computed meshes. Our method smartly incorporates useful rigid protein docking priors including commutativity and pairwise independent SE(3)-equivariances, thus avoiding the computational burden of data augmentation.

We look forward to incorporating more domain knowledge in EQUIDOCK and extend it for flexible docking and docking molecular dynamics, as well as adapt it to other related tasks such as drug binding prediction. On the long term, we envision that fast and accurate deep learning models would allow us to tackle more complex and involved biological scenarios, for example to model the mechanism of action of various drugs or to design de novo binding proteins and drugs to specific targets (e.g. for antibody generation). Last, we hope that our architecture can inspire the design of other types of biological 3D interactions.

**Limitations.** First, our presented model does not incorporate protein flexibility which is necessary for various protein families, e.g., antibodies. Unfortunately, both DB5 and DIPS datasets are biased towards rigid body docking . Second, we only prevent steric clashes using a soft constraint (Eq. (15)) which has limitations (see Table 6). Future extensions would hard-constrain the model to prevent such artifacts.

## ACKNOWLEDGEMENTS

The authors thank Hannes Stärk, Gabriele Corso, Patrick Walters, Tian Xie, Xiang Fu, Jacob Stern, Jason Yim, Lewis Martin, Jeremy Wohlwend, Jiaxiang Wu, Wei Liu, and Ding Xue for insightful and helpful discussions. OEG is funded by the Machine Learning for Pharmaceutical Discovery and Synthesis (MLPDS) consortium, the Abdul Latif Jameel Clinic for Machine Learning in Health, the DTRA Discovery of Medical Countermeasures Against New and Emerging (DOMANE) threats program, and the DARPA Accelerated Molecular Discovery program. This publication was created as part of NCCR Catalysis (grant number 180544), a National Centres of Competence in Research funded by the Swiss National Science Foundation. RB and TJ also acknowledge support from NSF Expeditions grant (award 1918839): Collaborative Research: Understanding the World Through Code.

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

# Appendix

C ONTENTS

## A    R EPRESENTING P ROTEINS AS G RAPHS

A protein is comprised of amino acid residues. The structure of an amino acid residue is shown in
Figure Fig. 7. Generally, an amino acid residue contains amino (-NH-), $\alpha$-carbon atom and carboxyl
(-CO-), along with a side chain (R) connected with the $\alpha$-carbon atom. The side chain (R) is specific
to each type of amino acid residues.

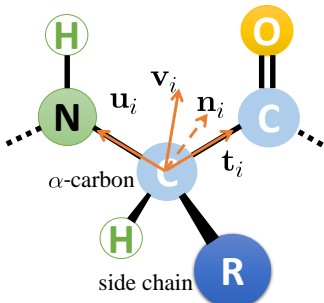

Figure 7: Representation of an amino acid residue and its local coordinate system.

We work on residue level (our approaches can be extended to atom level as well).  A protein is
represented by a set of nodes where each node is an amino acid residue in the protein. Each node $i$
has a 3D coordinate $\mathbf{x}_i \in \mathbb{R}^3$ which is the 3D coordinate of $\alpha$-carbon atom of the residue.

The neighborhood of a node is the set of $k$ ($k = 10$ in our experiments) nearest nodes where the
distance is the Euclidean distance between 3D coordinates.

Node feature is a one dimension indicator (one-hot encoding) of the type of amino acid residue. This
one dimension indicator will be passed into an embedding layer.

**Local Coordinate System.**    Similar to Ingraham et al. (2019) and Jumper et al. (2021), we introduce
a local coordinate system for each residue which denotes the orientation of a residue.  Based on
this, we can further design SE(3)-invariant edge features. As shown in Figure 7, for a residue $i$, we

denote the unit vector pointing from $\alpha$-carbon atom to nitrogen atom as $\mathbf{u}_i$. We denote the unit vector pointing from $\alpha$-carbon atom to carbon atom of the carboxyl (-CO-) as $\mathbf{t}_i$. $\mathbf{u}_i$ and $\mathbf{t}_i$ together define a plane, and the normal of this plane is $\mathbf{n}_i = \frac{\mathbf{u}_i \times \mathbf{t}_i}{\|\mathbf{u}_i \times \mathbf{t}_i\|}$. Finally, we define $\mathbf{v}_i = \mathbf{n}_i \times \mathbf{u}_i$. Then $\mathbf{n}_i$, $\mathbf{u}_i$ and $\mathbf{v}_i$ together form the basis of residue $i$'s local coordinate system. They together encode the orientation of residue $i$.

Then we introduce the edge features of an edge $j \rightarrow i \in \mathcal{E}$. These features describe the *relative position* of $j$ with respect to $i$, the *relative orientation* of $j$ with respect to $i$ and the *distance* between $j$ and $i$.

**Relative Position Edge Features**    First we introduce the edge features $\mathbf{p}_{j \rightarrow i}$ which describe relative position of $j$ with respect to $i$:

$$\mathbf{p}_{j \rightarrow i} = \begin{bmatrix} \mathbf{n}_i^\top \\ \mathbf{u}_i^\top \\ \mathbf{v}_i^\top \end{bmatrix} [\mathbf{x}_j - \mathbf{x}_i]$$

**Relative Orientation Edge Features**    As we mention above, each residue has orientation which carries information. Here we introduce the edge features $\mathbf{q}_{j \rightarrow i}$, $\mathbf{k}_{j \rightarrow i}$ and $\mathbf{t}_{j \rightarrow i}$ which describe relative orientation of $j$ with respect to $i$:

$$\mathbf{q}_{j \rightarrow i} = \begin{bmatrix} \mathbf{n}_i^\top \\ \mathbf{u}_i^\top \\ \mathbf{v}_i^\top \end{bmatrix} [\mathbf{n}_j], \quad \mathbf{k}_{j \rightarrow i} = \begin{bmatrix} \mathbf{n}_i^\top \\ \mathbf{u}_i^\top \\ \mathbf{v}_i^\top \end{bmatrix} [\mathbf{u}_j], \quad \mathbf{t}_{j \rightarrow i} = \begin{bmatrix} \mathbf{n}_i^\top \\ \mathbf{u}_i^\top \\ \mathbf{v}_i^\top \end{bmatrix} [\mathbf{v}_j]$$

**Distance-Based Edge Features**    Distance also carries information. Here we use radial basis function of distance as edge features:

$$\mathbf{f}_{j \rightarrow i, r} = e^{-\frac{(\|\mathbf{x}_j - \mathbf{x}_i\|)^2}{2\sigma_r^2}}, r = 1, 2, ..., R$$

Where $R$ and scale parameters $\{\sigma_r\}_{1 \leq r \leq R}$ are hyperparameters. In experiments, the set of scale parameters we used is $\{1.5^x | x = 0, 1, 2, ..., 14\}$. So for each edge, there are 15 distance-based edge features.

**Surface Aware Node Features**    We additionally compute 5 surface aware node features defined in Eq. (16) using $\lambda \in \{1., 2., 5., 10., 30.\}$.

# B    PROOFS OF THE MAIN PROPOSITIONS

## B.1    PROOF OF PROPOSITION 1.

*Proof.* Denote the predicted ligand position by $\mathbf{R}(\mathbf{X}_1|\mathbf{X}_2)\mathbf{X}_1 + \mathbf{t}(\mathbf{X}_1|\mathbf{X}_2) = \tilde{\mathbf{X}}_1$.

Assume first that SE(3)-invariance of the predicted docked complex defined by Eq. (1) is satisfied. Then the transformation to dock $\mathbf{Q}_1\mathbf{X}_1 + \mathbf{g}_1$ with respect to $\mathbf{Q}_2\mathbf{X}_2 + \mathbf{g}_2$ is the same as the transformation to change $\mathbf{Q}_1\mathbf{X}_1 + \mathbf{g}_1$ into $\mathbf{Q}_2\tilde{\mathbf{X}}_1 + \mathbf{g}_2$. We use the notation: $\mathbf{R}^\top(\mathbf{X}_1|\mathbf{X}_2) = (\mathbf{R}(\mathbf{X}_1|\mathbf{X}_2))^\top$. Then, we have the following derivation steps:

$\mathbf{R}(\mathbf{X}_1|\mathbf{X}_2)\mathbf{X}_1 + \mathbf{t}(\mathbf{X}_1|\mathbf{X}_2) = \tilde{\mathbf{X}}_1$

$\mathbf{X}_1 + \mathbf{R}^\top(\mathbf{X}_1|\mathbf{X}_2)\mathbf{t}(\mathbf{X}_1|\mathbf{X}_2) = \mathbf{R}^\top(\mathbf{X}_1|\mathbf{X}_2)\tilde{\mathbf{X}}_1$

$\mathbf{X}_1 + \mathbf{R}^\top(\mathbf{X}_1|\mathbf{X}_2)\mathbf{t}(\mathbf{X}_1|\mathbf{X}_2) = \mathbf{R}^\top(\mathbf{X}_1|\mathbf{X}_2)\mathbf{Q}_2^\top(\mathbf{Q}_2\tilde{\mathbf{X}}_1 + \mathbf{g}_2 - \mathbf{g}_2)$

$\mathbf{X}_1 + \mathbf{R}^\top(\mathbf{X}_1|\mathbf{X}_2)\mathbf{t}(\mathbf{X}_1|\mathbf{X}_2) = \mathbf{R}^\top(\mathbf{X}_1|\mathbf{X}_2)\mathbf{Q}_2^\top(\mathbf{Q}_2\tilde{\mathbf{X}}_1 + \mathbf{g}_2) - \mathbf{R}^\top(\mathbf{X}_1|\mathbf{X}_2)\mathbf{Q}_2^\top\mathbf{g}_2$

$\mathbf{X}_1 + \mathbf{R}^\top(\mathbf{X}_1|\mathbf{X}_2)\mathbf{t}(\mathbf{X}_1|\mathbf{X}_2) + \mathbf{R}^\top(\mathbf{X}_1|\mathbf{X}_2)\mathbf{Q}_2^\top\mathbf{g}_2 = \mathbf{R}^\top(\mathbf{X}_1|\mathbf{X}_2)\mathbf{Q}_2^\top(\mathbf{Q}_2\tilde{\mathbf{X}}_1 + \mathbf{g}_2)$

$\mathbf{Q}_1^\top(\mathbf{Q}_1\mathbf{X}_1 + \mathbf{g}_1 - \mathbf{g}_1) + \mathbf{R}^\top(\mathbf{X}_1|\mathbf{X}_2)(\mathbf{t}(\mathbf{X}_1|\mathbf{X}_2) + \mathbf{Q}_2^\top\mathbf{g}_2) = \mathbf{R}^\top(\mathbf{X}_1|\mathbf{X}_2)\mathbf{Q}_2^\top(\mathbf{Q}_2\tilde{\mathbf{X}}_1 + \mathbf{g}_2)$

$\mathbf{Q}_1^\top(\mathbf{Q}_1\mathbf{X}_1 + \mathbf{g}_1) - \mathbf{Q}_1^\top\mathbf{g}_1 + \mathbf{R}^\top(\mathbf{X}_1|\mathbf{X}_2)(\mathbf{t}(\mathbf{X}_1|\mathbf{X}_2) + \mathbf{Q}_2^\top\mathbf{g}_2) = \mathbf{R}^\top(\mathbf{X}_1|\mathbf{X}_2)\mathbf{Q}_2^\top(\mathbf{Q}_2\tilde{\mathbf{X}}_1 + \mathbf{g}_2)$

$\mathbf{R}(\mathbf{X}_1|\mathbf{X}_2)\mathbf{Q}_1^\top(\mathbf{Q}_1\mathbf{X}_1 + \mathbf{g}_1) - \mathbf{R}(\mathbf{X}_1|\mathbf{X}_2)\mathbf{Q}_1^\top\mathbf{g}_1 + \mathbf{t}(\mathbf{X}_1|\mathbf{X}_2) + \mathbf{Q}_2^\top\mathbf{g}_2 = \mathbf{Q}_2^\top(\mathbf{Q}_2\tilde{\mathbf{X}}_1 + \mathbf{g}_2)$

$\mathbf{Q}_2\mathbf{R}(\mathbf{X}_1|\mathbf{X}_2)\mathbf{Q}_1^\top(\mathbf{Q}_1\mathbf{X}_1 + \mathbf{g}_1) - \mathbf{Q}_2\mathbf{R}(\mathbf{X}_1|\mathbf{X}_2)\mathbf{Q}_1^\top\mathbf{g}_1 + \mathbf{Q}_2\mathbf{t}(\mathbf{X}_1|\mathbf{X}_2) + \mathbf{g}_2 = \mathbf{Q}_2\tilde{\mathbf{X}}_1 + \mathbf{g}_2$

From the last equation above, one derives the transformation of $\mathbf{Q}_1\mathbf{X}_1 + \mathbf{g}_1$ into $\mathbf{Q}_2\tilde{\mathbf{X}}_1 + \mathbf{g}_2$, which is, by definition of the functions $\mathbf{R}$ and $\mathbf{t}$, the same as the transformation to dock $\mathbf{Q}_1\mathbf{X}_1 + \mathbf{g}_1$ with respect to $\mathbf{Q}_2\mathbf{X}_2 + \mathbf{g}_2$. This transformation is

$$\mathbf{R}(\mathbf{Q}_1\mathbf{X}_1 + \mathbf{g}_1 | \mathbf{Q}_2\mathbf{X}_2 + \mathbf{g}_2) = \mathbf{Q}_2\mathbf{R}(\mathbf{X}_1|\mathbf{X}_2)\mathbf{Q}_1^\top$$

$$\mathbf{t}(\mathbf{Q}_1\mathbf{X}_1 + \mathbf{g}_1 | \mathbf{Q}_2\mathbf{X}_2 + \mathbf{g}_2) = \mathbf{Q}_2\mathbf{t}(\mathbf{X}_1|\mathbf{X}_2) - \mathbf{Q}_2\mathbf{R}(\mathbf{X}_1|\mathbf{X}_2)\mathbf{Q}_1^\top\mathbf{g}_1 + \mathbf{g}_2$$

which concludes the proof.

Conversely, assuming constraints in Eq. (2) hold, we derive that $\mathbf{Q}_1\mathbf{X}_1 + \mathbf{g}_1$ is transformed into $\mathbf{Q}_2\tilde{\mathbf{X}}_1 + \mathbf{g}_2$, which then is trivial to check that it satisfies SE(3)-invariance of the predicted docked complex defined by Eq. (1).

□

## B.2 PROOF OF PROPOSITION 3.

*Proof.* We use the notation $\mathbf{R}^\top(\mathbf{X}_1|\mathbf{X}_2) := (\mathbf{R}(\mathbf{X}_1|\mathbf{X}_2))^\top$. As in Appendix B.1, we denote $\mathbf{R}(\mathbf{X}_1|\mathbf{X}_2)\mathbf{X}_1 + \mathbf{t}(\mathbf{X}_1|\mathbf{X}_2) = \tilde{\mathbf{X}}_1$. Then the transformation to dock $\mathbf{X}_2$ with respect to $\mathbf{X}_1$ is the same as the transformation to change $\tilde{\mathbf{X}}_1$ back to $\mathbf{X}_1$, which is

$$\mathbf{R}(\mathbf{X}_1|\mathbf{X}_2)\mathbf{X}_1 + \mathbf{t}(\mathbf{X}_1|\mathbf{X}_2) = \tilde{\mathbf{X}}_1$$

$$\mathbf{X}_1 + \mathbf{R}^\top(\mathbf{X}_1|\mathbf{X}_2)\mathbf{t}(\mathbf{X}_1|\mathbf{X}_2) = \mathbf{R}^\top(\mathbf{X}_1|\mathbf{X}_2)\tilde{\mathbf{X}}_1$$

$$\mathbf{X}_1 = \mathbf{R}^\top(\mathbf{X}_1|\mathbf{X}_2)\tilde{\mathbf{X}}_1 - \mathbf{R}^\top(\mathbf{X}_1|\mathbf{X}_2)\mathbf{t}(\mathbf{X}_1|\mathbf{X}_2)$$

From the last equation above, we derive the transformation to change $\tilde{\mathbf{X}}_1$ back to $\mathbf{X}_1$, which is the same as the transformation to dock $\mathbf{X}_2$ with respect to $\mathbf{X}_1$. □

## B.3 PROOF OF PROPOSITION 4.

*Proof.* Let $\mathbf{X}_1^{(l+1)}, \mathbf{H}_1^{(l+1)}, \mathbf{X}_2^{(l+1)}, \mathbf{H}_2^{(l+1)} = \text{IEGMN}(\mathbf{X}_1^{(l)}, \mathbf{H}_1^{(l)}, \mathbf{X}_2^{(l)}, \mathbf{H}_2^{(l)})$ be the output of an IEGMN layer. Then, for any matrices $\mathbf{Q}_1, \mathbf{Q}_2 \in SO(3)$ and any translation vectors $\mathbf{g}_1, \mathbf{g}_2 \in \mathbb{R}^3$, we want to prove that IEGMN satisfy the pairwise independent SE(3)-equivariance property:

$$\mathbf{Q}_1\mathbf{X}_1^{(l+1)} + \mathbf{g}_1, \mathbf{H}_1^{(l+1)}, \mathbf{Q}_2\mathbf{X}_2^{(l+1)} + \mathbf{g}_2, \mathbf{H}_2^{(l+1)} = \text{IEGMN}(\mathbf{Q}_1\mathbf{X}_1^{(l)} + \mathbf{g}_1, \mathbf{H}_1^{(l)}, \mathbf{Q}_2\mathbf{X}_2^{(l)} + \mathbf{g}_2, \mathbf{H}_2^{(l)})$$

where each column of $\mathbf{X}_1^{(l)} \in \mathbb{R}^{3 \times n}, \mathbf{H}_1^{(l)} \in \mathbb{R}^{d \times n}, \mathbf{X}_2^{(l)} \in \mathbb{R}^{3 \times m}$ and $\mathbf{H}_2^{(l)} \in \mathbb{R}^{d \times m}$ represent an individual node's coordinate embedding or feature embedding.

We first note that the equations of our proposed IEGMN layer that compute messages $\mathbf{m}_{j \to i}, \mu_{j \to i}$, $\mathbf{m}_i$ and $\mu_i$ are SE(3)-invariant. Indeed, they depend on the initial features which are SE(3)-invariant by design, the current latent node embeddings $\{\mathbf{h}_i^{(l)}\}_{i \in \mathcal{V}_1 \cup \mathcal{V}_2}$, as well as the Euclidean distances on the current node coordinates $\{\mathbf{x}_i^{(l)}\}_{i \in \mathcal{V}_1 \cup \mathcal{V}_2}$. Thus, we also derive that the equation that computes the new latent node embeddings $\mathbf{h}_i^{(l+1)}$ is SE(3)-invariant. Last, the equation that updates the coordinates $\mathbf{x}_i^{(l+1)}$ is SE(3)-equivariant with respect to the 3D coordinates of nodes from the same graph as $i$, but SE(3)-invariant with respect to the 3D coordinates of nodes from the other graph since it only uses invariant transformations of the latter.

□

# C SURFACE FEATURES

**Visualization.** We further discuss our new surface features introduced in Eq. (16). We first visualize their design intuition in Fig. 8. A synthetic experiment is shown in Fig. 9.

**Correlation with MSMS features.** Next, we analyze how accurate are these features compared to established residue depth estimation methods, e.g. based on the MSMS software (Sanner et al., 1996). We plot the Spearman rank-order correlation of the two methods in Fig. 10. We observe a concentrated distribution with a mean of 0.68 and a median of 0.70, suggesting a strong correlation with the MSMS depth estimation.

**Closed form expression.** Finally, we prove that for points close to the protein surface and surrounded by (infinitely) many equally-distanced and equally-spaced points, one can derive a closed form expression of the surface features defined in Eq. (16). See Fig. 11. We work in 2 dimensions, but extensions to 3 dimensions are straightforward. Assume that the local surface at point $\mathbf{x}_i$ has angle $\alpha$. Further, assume that $\mathbf{x}_i$ is surrounded by $N$ equally-distanced and equally-spaced points denoted by $\mathbf{x}'_i$. Then, all $w_{i,i',\lambda}$ will be identical. Then, the summation vector in the numerator of Eq. (16) will only have non-zero components on the direction that bisects the surface angle, as the other components will cancel-out. Then, under the limit $N \to \infty$, we derive the closed form expression:

$$\rho_i(\mathbf{x}_i; \lambda) = \frac{1}{N} \left\| \sum_{i' \in \mathcal{N}_i} \frac{\mathbf{x}_i - \mathbf{x}_{i'}}{\|\mathbf{x}_i - \mathbf{x}_{i'}\|} \right\| = \frac{2}{N} \sum_{j=0}^{\frac{N}{2}} \cos(\frac{j\alpha}{N}) \approx_{N \to \infty} \frac{2}{\alpha} \int_0^{\alpha/2} \cos(\theta) d\theta = 2\frac{\sin(\alpha/2)}{\alpha} \tag{17}$$

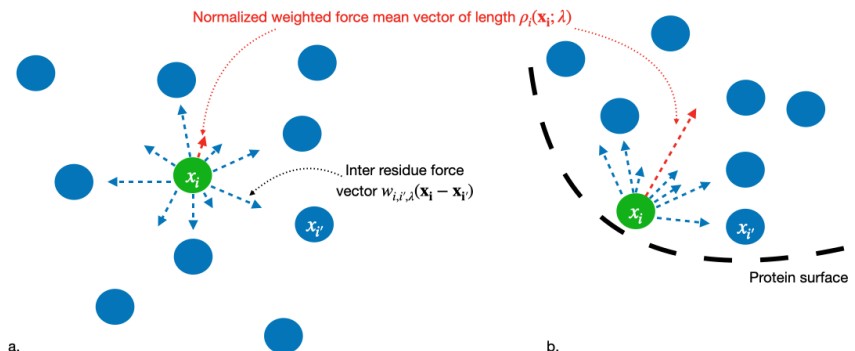

Figure 8: Intuition behind surface features defined in Eq. (16). **a.** Residues in the core (interior) of a protein are likely to have a small weighted average of directionally spread neighboring forces, while **b.** residues close to the surface receive vector contributions from a narrower space subset and, thus, have larger $\rho$ feature values.

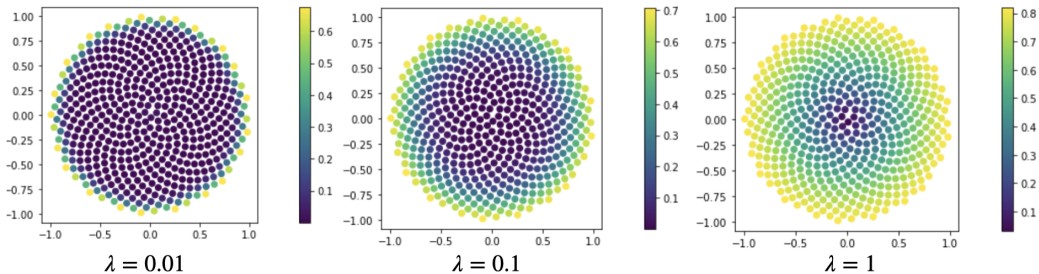

Figure 9: Distribution of our surface feature values defined in Eq. (16) for 500 points uniformly distributed in the unit circle. One can notice a strong correlation with the depth (i.e. distance to surface) which is further quantified in Fig. 10. Note that the scale for $\lambda$ in this synthetic experiment differs from that of real proteins.

## D DATASETS

The overview of datasets is in Table 2. DB5.5 is obtained from https://zlab.umassmed.edu/benchmark/, while DIPS is downloaded from https://github.com/drorlab/DIPS. While DIPS contains only the bound structures, thus currently being only suitable for rigid docking, DB5.5 also includes unbound protein structures, however, mostly showing rigid structures - see Fig. 12.

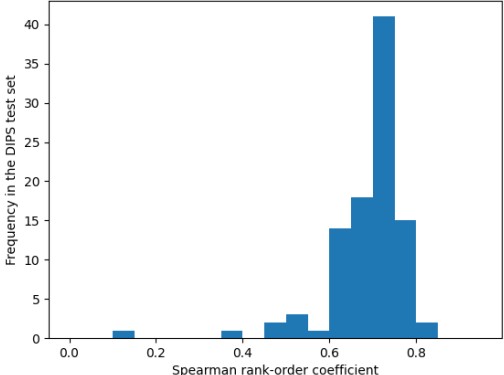

Figure 10: Distribution of the Spearman rank-order coefficient computed per each protein as the correlation between MSMS residues' depths and our surface features defined in Eq. (16) (for $\lambda = 30$). Histogram computed over the ligands in the DIPS test set (100 proteins).

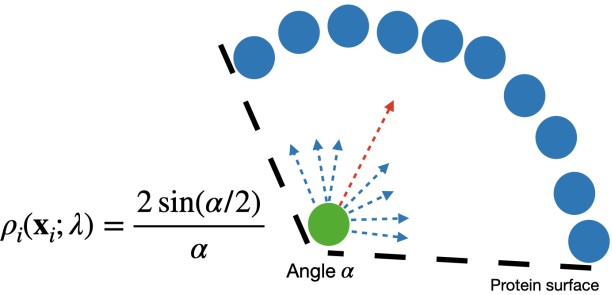

$$\rho_i(\mathbf{x}_i; \lambda) = \frac{2\sin(\alpha/2)}{\alpha}$$

Figure 11: For points close to the protein surface where the local surface angle is $\alpha$ we can derive a closed form expression for the surface feature defined in Eq. (16) under the assumption of being surrounded by infinitely many points at approximately equal distances and equally-spaced . A similar derivation is possible in 3D.

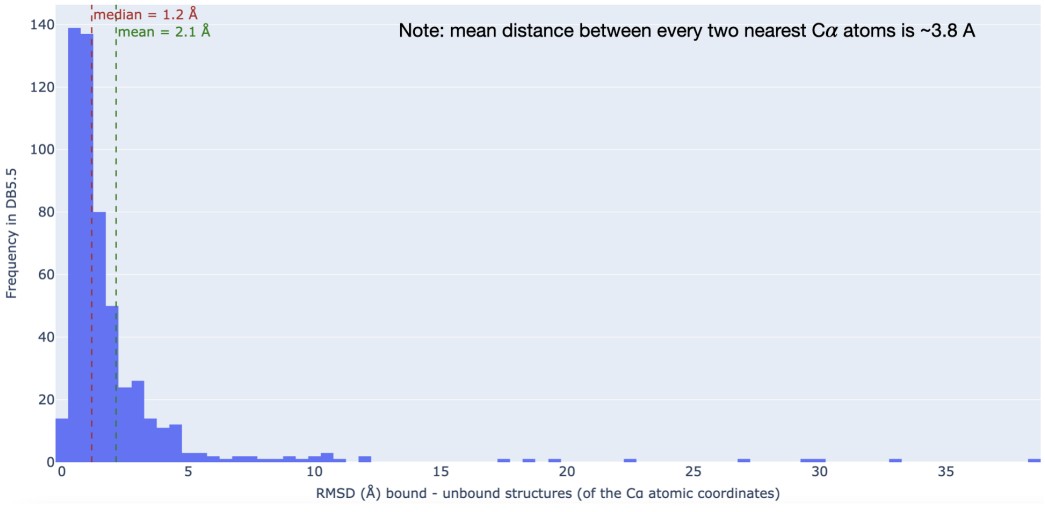

Figure 12: Distance (RMSD) between unbound and bound structures of the DB5.5 dataset reveals that most of the proteins are relatively rigid. Thus, better datasets are needed to tackle the docking conformational change problem.

Table 2: **Overview of Datasets.** For DIPS, the statistics of number of residues and atoms per protein is based on a subset consisting of 200 proteins.

| Dataset | # Pairs of Proteins | # Residues per Protein | # Atoms per Protein |
|---|---|---|---|
| DIPS | 41876 | 276 ($\pm$189) | 2159 ($\pm$1495) |
| DB5.5 | 253 | 268 ($\pm$215) | 2089 ($\pm$1694) |

## E  MORE EXPERIMENTAL DETAILS AND RESULTS

**Baseline Failures.**  On the test sets, ATTRACT fails for '1N2C' in DB5.5, 'oi_4oip.pdb1_8', 'oi_4oip.pdb1_3' and 'p7_4p7s.pdb1_2' in DIPS. For such failure cases, we use the unbound input structure as the prediction for metrics calculation.

**Hyperparameters.**  We perform hyperparameter search over the choices listed in Table 3 and select the best hyperparameters for DB5.5 and DIPS respectively based on their corresponding validation sets.

Table 3: Hyperparameter choices. LN stands for layer normalization, BN stands for batch normalization.

| Hyperparameter | Choice |
|---|---|
| Node degree (for k-NN) | 10 |
| Weight of the intersection loss | 0.0, 1.0 |
| Normalization for $h_i$ of IEGMN layers | No, LN |
| Normalization for others | No, BN, LN |
| Number of attention heads ($K$) | 25, 50, 100 |
| Slope of leaky relu | 0.1, 0.01 |
| Dimension of $h_i$ of IEGMN layers | 32, 64 |
| Dimension of residue type embedding | 32, 64 |
| Number of IEGMN layers | 5, 8 |
| If IEGMN layers except the first one share parameters | True, False |
| $\eta$ of coordinates skip connection | 0.0, 0.25 |
| Weight decay | 0, 1e-5, 1e-4, 1e-3 |

**Detailed Running Times.**  In addition to the main text, we show in Table 4 detailed running times of all methods. Hardware specifications are as follows: ATTRACT was run on a 6-Core Intel Core i7 2.2 GHz CPU; HDOCK was run on a single Intel Xeon Gold 6230 2.1 GHz CPU; EQUIDOCK was run on a single Intel Core i9-9880H 2.3 GHz CPU. CLUSPRO and PATCHDOCK have been manually run using their respective web servers.

**Plots for DB5.5.**  We show the corresponding plots for DB5.5 results in Fig. 13.

Table 4: Inference time comparison (in seconds). Note: ClusPro and PatchDock were run manually using the respective public webservers, thus their runtimes are influenced by their cluster load.

| Methods | Runtime on DIPS Test Set | | | | | Runtime on DB5.5 Test Set | | | | |
|---|---|---|---|---|---|---|---|---|---|---|
| | Mean | Median | Min | Max | Std | Mean | Median | Min | Max | Std |
| ATTRACT (LOCAL) | 1285 | 793 | 62 | 8192 | 793 | 570 | 524 | 180 | 1708 | 373 |
| HDOCK (LOCAL) | 778 | 635 | 145 | 3177 | 570 | 615 | 461 | 210 | 2593 | 459 |
| CLUSPRO (WEB) | 10475 | 9831 | 2632 | 22654 | 4512 | 15507 | 14393 | 9207 | 28528 | 4126 |
| PATCHDOCK (WEB) | 7378 | 6900 | 600 | 16560 | 3979 | 3290 | 2820 | 1080 | 14520 | 2459 |
| **EQUIDOCK (LOCAL)** | **5** | **3** | **1** | **22** | **5** | **5** | **3** | **1** | **53** | **10** |

**Ablation Studies.** To highlight contributions of different model components, we provide ablation studies in Table 5. One can note that, as expected, removing the pocket loss results in lower interface RMSD scores compared to removing other components.

**Analysis of the Intersection Loss.** We further analyze the intersection loss introduced in Eq. (15) with parameters $\gamma = 10$ and $\sigma = 25$ (chosen on DB5 validation set). We show in Table 6 that this loss achieves almost perfect values for the ground truth structures, being important to softly constrain non-intersecting predicted proteins.

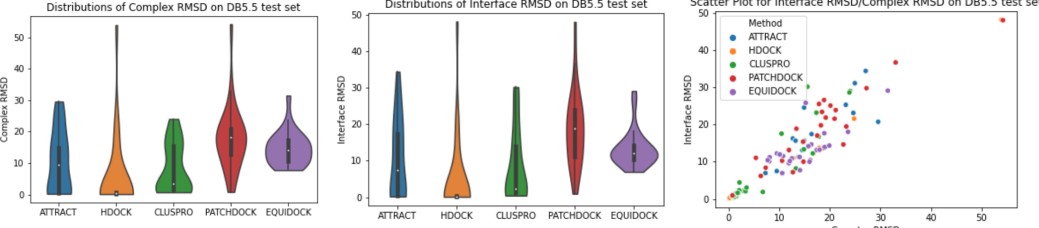

Figure 13: DB5.5 test results: **a.** Complex-RMSD distributions; **b.** Interface-RMSD distributions; **c.** scatter plot for C-RMSD vs I-RMSD.

Table 5: Ablation studies. We show DIPS test median C-RMSD and I-RMSD values for the corresponding best validation models. Abbreviations: "intersection loss" = intersection loss in Eq. (15), "pocket loss" = pocket loss in Eq. (14), "surface feas" = surface features in Eq. (16).

| Model | C-RMSD | I-RMSD |
|---|---|---|
| Full model | 13.29 | 10.18 |
|     without pocket loss | 15.91 | 12.01 |
|     without pocket loss, intersection loss | 16.43 | 12.92 |
|     without pocket loss, surface feas | 14.80 | 13.10 |
|     without pocket loss, intersection loss, surface feas | 15.19 | 11.38 |
|     without surface feas | 13.73 | 10.65 |
|     without intersection loss | 15.49 | 11.09 |
|     without intersection loss, surface feas | 15.04 | 10.94 |

Table 6: Values of the intersection loss defined in Eq. (15) and evaluated on the DIPS validation set in different scenarios. "Centered structures" means that both ground truth ligand and receptor point clouds have been centered (0-mean), without any other modifications.

| Centered structures | EquiDock trained with intersection loss | EquiDock trained without intersection loss | Ground truth complexes |
|---|---|---|---|
| 56.42 | 12.68 | 21.03 | 1.16 |

