# OpenReview forum: "Independent SE(3)-Equivariant Models for End-to-End Rigid Protein Docking"
_ICLR.cc/2022/Conference — ICLR 2022 Spotlight_

### Official Review · Reviewer_ydqb · 2021-11-02

**Correctness:** 4
**Technical Novelty And Significance:** 3
**Empirical Novelty And Significance:** 3
**Recommendation:** 8
**Confidence:** 3

**Main Review:**

 Pros:
The whole paper is constructed and written very well.
The authors nicely show the proof of SE(3)-invariant and commutativity for the rigid body docking problem.
Following the invariance property, the paper introduces the novel graph matching network and avoids sampling step based on it.
Using a soft loss function avoids point cloud intersection.

Cons:
As much as we appreciated the approach presented here the performance just seems unacceptable compared to other methods
The author uses two baselines which were developed 4 years ago. The reason they didn’t use other methods, no local packages, is somewhat problematic too. Still, their performance is much worse than one of the methods and hypothesizing the test data was used to train those models does not resolve this issue.
Their runtime is much faster. However, HDOCK, which is the best performance model among those three, uses a bearable amount of time and achieves much better performance.
Three models are ran on different hardware, which makes runtime comparison problematic.

Lack of result analysis
It may have been helpful to show the method proposed can provide comparable results on more downstream tasks. Even though its rigid docking prediction is not good enough, maybe the prediction has other properties which can help the other perspectives.

We also noted the method uses soft constraints to discourage point cloud intersections. However, the authors do not show/discuss if it really prevents this from happening and why not use hard constraints directly (stability during training?).

Other Comments:
It is not clear how the number of keypoints K was chosen in the “Keypoints for Differentiable Protein Superimposition” section.

Also we noticed that the whole section of proofs for propositions lacks citations but they are not claimed as novelty at the beginning. Are they part of the contributions or not? How does these parts relate to previous works?

Update:
===========
The improvements implemented in the revised manuscript and clarifications supplied by the authors elevated many of the above concerns and we therefore updated the evaluation to reflect this.

**Summary Of The Paper:**

This paper proposes an end-to-end deep learning architecture to model the rigid body protein docking problem. By incorporating the inductive biases of SE(3)-invariance of the final docking position and commutativity, the proposed method avoids the millions of sampling and achieves a competitive performance with much faster speed. In addition, they discover and align keypoints by the attention-based selection algorithm and use optimal transport to predict the binding pocket location based on those selected points. The main contribution of this paper is the combination of the novel graph matching networks and keypoint selection algorithm to predict the 3D position of the docking model. Results are shown on the task of 1) Protein docking complex prediction 2) Runtime

**Summary Of The Review:**

Overall we found the paper to be clearly written and offer a nice formulation with a matching modeling approach for the important problem of rigid body protein docking. In that sense we definitely liked the work. However, the results are just unimpressive and no other insights/utility are offered beyond improved running time compared to standard techniques available.

---

> ### Author Response · Authors · 2021-11-15
> **Thank you. Our updates and comments**
>
> We thank the reviewer for the extensive feedback that helps us to improve our work. Please see our comments below.
>
> &nbsp;
>
> - **Performance and improved model quality:** As we have commented in the main message addressed to all reviewers, we have **significantly improved numbers** and **two more established and popular baselines**, that we now **often outperform**. Please see our updated Table 1.
>
> &nbsp;
>
> - **Importance of our dramatic runtime improvements:** Please see the main message addressed to all reviewers where we argue on multiple different perspectives.
>
> &nbsp;
>
> - **Novelty of our theory:** All propositions in Section 3 are our novel theoretical contributions. Moreover, formalizing the notion of pairwise independent Euclidean-equivariance for 3D object interaction problems is **our novel contribution**. We have commented on this aspect in the main message addressed to all reviewers. We have rewritten the Contribution and Related Work sections to clarify this aspect.
>
> &nbsp;
>
> - **Other merits and applications of our model:** As we have commented in the main message addressed to all reviewers, our model can be applied to other tasks, and, in fact, we have already obtained improvements over recent state-of-the-art baselines on drug-target binding. Moreover, our model has other advantages over the baselines such as significant speed-ups or wider applicability. Please refer to the main message.
>
> &nbsp;
>
> - **Soft constraints for point cloud intersections:** We added **new results in Table 6** (in Appendix E) to support the utility of our intersection loss in preventing point cloud intersections. Notice that our proposed intersection loss achieves near perfect values for the ground truth structures, being a very efficient alternative to more expensive intersection criteria, such as based on the MSMS software.
>
> &nbsp;
>
> - **Hard constraints for point cloud intersections:** Hard constraints such as projections during gradient descent updates make training and optimization particularly difficult and unstable. We use soft-constraints to regularize our model to prevent body intersections.
>
> &nbsp;
>
> - **Other model insights:** Please see the paragraph called “How accurate are our surface features?” from our main message.
>
> &nbsp;
>
> - **Different hardware for running the baselines:** Unfortunately, these proprietary software can only be run on specific operating systems with specific configurations. Moreover, for the web server baselines (ClusPro and PatchDock), we had to manually submit each test pair individually.
>
> &nbsp;
>
> - **Choosing the number of keypoints K:** We do a hyperparameter search of K over the values 25, 50 and 100. We find that K=50 works best for both DIPS and DB5 datasets. Moreover, there exist, on average, 40 binding pocket point pairs per each protein in the train set (binding pocket points are defined for the threshold of 8 A), which hints that K=50 is a reasonable choice.

---

> > ### Comment · Reviewer_ydqb · 2021-11-19
> > **Response to revision**
> >
> > We thank the authors for their detailed response. We think the authors did a good job with the revision improving performance, adding analysis, and clarifying design and parameter choice. While the overall performance still lags compared to HDock the authors note it is unclear whether that may be due to the latter being trained on the same data. Nonetheless, with the additional analysis and clarification about the specific contributions of this work we believe there is a strong case for this work and will be updating our score accordingly. We congratulate the authors on a job well done addressing our and the other reviewers' comments.

---

> > > ### Author Response · Authors · 2021-11-20
> > > **Thank you!**
> > >
> > > Thank you for your valuable feedback and appreciation ! Best regards.

---

### Official Review · Reviewer_VRJP · 2021-11-02

**Correctness:** 4
**Technical Novelty And Significance:** 3
**Empirical Novelty And Significance:** 2
**Recommendation:** 8
**Confidence:** 5

**Main Review:**

The authors have put a significant amount of time into designing, developing, and testing their proposed method, and it introduces and adapts several interesting ideas. In particular, the core contribution is their proposed transformation, which guarantees pairwise independent SE(3)-equivariance for the two sets of 3D coordinates. My comments are as follows:

- Since no conformational changes are allowed, the residues buried in the hydrophobic core of the proteins play no role in predicting the binding conformation. Calculating solvent-accessible surface area is one of the most well-studied problems in structural genomics. The authors could use one of such methods, e.g., one based on the Voronoi procedure [PMID: 12376381], to significantly reduce the problem size and render identifying keypoints easier.
- The authors have a loss term to avoid point cloud intersections, but there are no terms for preventing steric clashes by enforcing Van der Waals radii of proximal atoms. Have the authors checked for steric clashes in the binding pocket?
- Analyzing performance metrics needs more elaboration. For example, the distribution of C/I RMSDs could be plotted, and/or scatter plots could be added. Moreover, statistical tests could be performed to reliably show whether the proposed method outperforms the other two. The same applies to inference times and a box/violin plot might be a better option.
- Judging by the results in Table 1, one might infer that HDOCK performs better than IEGMN. It is true that IEGMN has significantly lower inference times, but in practice, experimentally determining protein structures is time-intensive (in the order of months) and costly, and run times in the order of minutes or even hours do not matter.
- It is good to see that the authors have used two different RMSD definitions; however, the authors should consider looking at CAPRI criteria (critical assessment of predicted interactions criteria) [PMID: 12784359] and standardize their evaluation method.
- The authors employ several ideas and their contribution to the final performance is not clear. For example, what would happen if the “surface-aware node features” are not used?

Some minor notes:
- SE(3)-equivariance and SE(3)-invariance are used interchangeably.
- In p6, what does “normalized keypoints” mean, zero-mean?


**Summary Of The Paper:**

In this paper, the authors propose a method for the “rigid body” docking of protein-protein complexes, i.e., a complex in which conformational changes in protein structures are not allowed. This method, called independent E(3)-equivariant graph matching networks (IEGMNs), finds the optimal rotation and translation to place proteins in a manner that the distances between residues in the binding site is minimized. A core feature of this method is SE(3) invariance, which means the optimal solution is invariant to the rotations and translations of the two proteins. This is achieved in an elaborate manner employing several ideas and technics:

- A k-NN graph is built for each protein using its CA coordinates and a set of additional features are also extracted
- Message-passing neural network (MPNN) is used for graph-matching
- Binding pocket residues or “keypoints” are identified
- A differentiable Kabsch model is used for superimposing is the “keypoints”
- Optimal transport is used for the alignment of residues in the binding pocket

This method, as reported by the authors, runs very fast and can identify the protein complex efficiently.


**Summary Of The Review:**

The proposed method is interesting, but it combines too many ideas together without quantifying their contribution. Moreover, performance evaluation and comparisons with the existing methods are lacking.

---

> ### Author Response · Authors · 2021-11-15
> **Thank you. Our updates**
>
> We thank the reviewer for the extensive feedback that helps us to improve our work. Please see our comments below.
>
> &nbsp;
>
> - **Performance and improved model quality:** As we have commented in the main message addressed to all reviewers, we have **improved numbers** and **two more popular baselines** that we **often outperform**. Please see our updated Table 1.
>
> &nbsp;
>
> - **New plots, empirical insights and ablations:** We provide new plots and ablations in our updated paper. Please see Figures 4, 5, 9, 10, 11, Appendix E, Appendix C, Table 5, Table 6, and our main message addressed to all reviewers.
>
>
> &nbsp;
>
> - **Importance of our dramatic runtime improvements:** Please see the main message addressed to all reviewers where we argue on multiple different perspectives for why faster deep learning docking methods are desirable.
>
> &nbsp;
>
> - **Using surface area:**  We tried the following experiment. We only kept residues that are close to the surface of the protein, and applied our EquiDock model on those. To compute the residue depth (distance to the surface), we attempted to use the MSMS program. However, this was prohibitively slow on the large DIPS size (preprocessing and training would not finish in time for this rebuttal). We instead adopted a faster alternative. We show in Figures 9 and 10 that our surface feature in Eq. 16 and the MSMS residue depth are highly correlated, thus, the former provides a much faster proxy for the latter. Next, we only kept “surface residues”, defined as residues for which the feature in Eq. 16 (with lambda = 30) was at least 0.5. On top of those, we applied the same EquiDock model. However, these trained models were consistently worse than our “all-residues” model. We hypothesize that this happens for multiple reasons. First, using only surface residues significantly increases the graph distances between the majority of pairs of nodes and, thus, message passing neural networks (MPNNs) would need significantly more layers to capture long range residue interactions. This is important because MPNNs have various known difficulties in using too many layers [16,17,18], and we have seen that going beyond 8 MPNN layers results in no model improvements, but in increased training times and memory. Second, our “all-residues” model already incorporates surface information in features defined in Eq. 16, as a consequence having the flexibility to leverage or ignore core residues. Third, our intersection loss defined in Eq. 15 is no longer applicable for “empty objects”, i.e., using only surface residues. Last, prior work on protein contact prediction has proven success with models using all residues alone [19, 20, 21, 22] or in combination with the surface area [23]. We propose to leave a more thorough investigation of this aspect for future work.
>
>
> &nbsp;
>
> - **Steric clashes:** We thank the reviewer for this suggestion. Our model could enforce the avoidance of steric clashes with a loss similar to our non-intersection loss in Eq. 15 (which we have analyzed empirically in Table 6). That would require us to work at atom level instead of residue level (our current setting). In fact, we have already tried to use all atoms in the proteins instead of their residues. However, we see a 10 times GPU memory increase and significant running time increase (more than 2 times), but did not see any empirical improvements compared to using only residues. We leave further thorough investigation for future work.
>
>
> &nbsp;
>
> - **SE(3)-equivariance and SE(3)-invariance are used interchangeably:** We updated our paper to correct and clarify this issue. We improved Section 3, Figure 2 and the contributions paragraph (Introduction).
>
>
> &nbsp;
>
> - Normalized keypoints: zero-mean.
>
> &nbsp;
>
>
> &nbsp;
>
>
> [16] On the bottleneck of graph neural networks and its practical implications, Alon and Yahav, 2020
>
> [17] Graph neural networks exponentially lose expressive power for node classification, K Oono, T Suzuki, 2019
>
> [18] Deep learning long-range information in undirected graphs with wave networks, Matlock et al., 2019
>
> [19] Protein Interface Prediction using Graph Convolutional Networks, Fout et al., 2017
>
> [20] End-to-End Learning on 3D Protein Structure for Interface Prediction, Townshend et al., 2019
>
> [21] Hierarchical, rotation-equivariant neural networks to select structural models of protein complexes, Eismann et al., 2020
>
> [22] Deep Learning of High-Order Interactions for Protein Interface Prediction, Liu et al., 2020
>
> [23] Deep graph learning of inter-protein contacts, Xie and Xu, 2021

---

> > ### Comment · Reviewer_VRJP · 2021-11-19
> > **Response to the revisions**
> >
> > I highly appreciate the authors significant effort on improving their method and addressing the reviewers' comments. Based on the responses, I am going to update my score. Having said that, while the idea is novel and the short run time facilitates more extensive hit screening, EquiDock is still not competitive and I strongly encourage the authors to pursuit this line of research and improve their method further. For example, a short molecular dynamic run can fix small steric clashes and fix side chain rotamers.

---

> > > ### Author Response · Authors · 2021-11-20
> > > **Thank you!**
> > >
> > > Thank you for your valuable feedback! We look forward to incorporating your other suggestions in a mathematically principled and empirically stronger extension of EquiDock.

---

### Official Review · Reviewer_ArSS · 2021-11-03

**Correctness:** 4
**Technical Novelty And Significance:** 3
**Empirical Novelty And Significance:** 3
**Recommendation:** 8
**Confidence:** 5

**Main Review:**

## Strength
The proposed method is quite novel. Built upon Graph Matching Networks [1] and E(3)-Equivariant Graph Neural Networks [2], they propose a SE(3) equivariant graph matching network. The keypoint selection with optimal transport and rotation prediction with Kabsch algorithm are also interesting to me. The proposed method achieves a promising speedup of 80-500x.
The proposed method seems general and can be applied to other rigid docking scenarios, i.e., predicting a rotation and translation matrix. The algorithm can potentially be adapted for protein-ligand docking, which is also an important task in computational chemistry.
The paper is well written and the mathematical formulation of SE(3) equivariant rigid protein docking is very nice. The authors also promise they will release the code and datasets after reviewing the process.

## Weakness
Although the proposed method achieves a significant speedup compared to previous docking software, it does not always outperform the baselines as acknowledged by the authors. It is also unclear how structure refinement, or templates can be combined with the current method to further improve the performance, which are used in many other protein structure prediction algorithms, e.g., alphafold2.
The DIPS dataset does not contain protein structures in their unbounded form. It is unclear how the authors tackle this bias in the dataset.
The authors include Surface Aware Node Features into their model. I agree that surface contact modeling is very important for protein docking. It is therefore interesting to see an ablation study over the features.

Reference
[1] Graph matching networks for learning the similarity of graph structured objects
[2] E(n)-equivariant graph neural networks.


**Summary Of The Paper:**

The paper proposes a SE(3) equivariant graph matching network for end-to-end rigid protein docking. They propose a novel optimal transport loss to approximate binding pocket and a differentiable Kabsch algorithm to predict the docking pose. They achieve significant running time improvements over existing protein docking software with competitive results, and do not rely on heavy sampling, structure refinement, or templates


**Summary Of The Review:**

I vote for accepting this manuscript for the reasons listed above. However, several important issues remain to be addressed.

---

> ### Author Response · Authors · 2021-11-15
> **Thank you. Our updates.**
>
> We thank the reviewer for the extensive feedback that helps us to improve our work. Please see our comments below.
>
> &nbsp;
>
> - **Performance and improved model quality:** As we have commented in the main message addressed to all reviewers, we have improved numbers and more baselines that we often outperform. Please see our updated Table 1.
>
> &nbsp;
>
> - **Structure refinement:** This is a natural extension of our model. In an initial experiment, we have added a single E(3)-equivariant network (3 layers) on the predicted protein complex and trained it end-to-end with our EquiDock model. This model is exactly the same as our EquiDock, but now 3D coordinates are also updated using inter-protein cross-coordinate messages. The intuition is that an E(3)-equivariant network would fine-tune the predicted complex in a principled geometric manner. However, our initial set of experiments revealed no improvements of this strategy. We leave further exploration for future work.
>
> &nbsp;
>
> - **Templates:** Previous docking baselines have often relied on such important information like motifs [12]. We agree that it is useful to integrate templates into our model, potentially inspired by AlphaFold 2. However, it might be both undesirable and expensive to do that, and our line of thinking follows recent work [13,14,15]. First, a template database might be overfitted to a specific train set and, thus, not generalize well to new docking patterns. Second, it might not cover exhaustively all possible (unknown) motifs. Third, it would make our models computationally prohibitive for structures with a large number of local patterns. We believe this important direction deserves a more thorough future investigation.
>
> &nbsp;
>
> - **DIPS bias:** Indeed, DIPS is biased towards rigid docking and our model is tailored to this specific setup. Flexible docking is an important facet of this problem, and one possible way to de-bias DIPS would be as follows: take protein sequences in DIPS and predict their unbound structures with an accurate model such as RoseTTAFold or AlphaFold 2. In this way, we obtain access to both unbound and bound structures for all protein pairs in DIPS. Future models could use this approach to perform flexible protein docking. We look forward to extending EquiDock to this setup, potentially combining it with ideas from AlphaFold 2 or RoseTTAFold.
>
> &nbsp;
>
> [12] PatchMAN docking: Modeling peptide-protein interactions in the context of the receptor surface, Khramushin et al., 2021
>
> [13] Fast end-to-end learning on protein surfaces, Sverrisson et al., 2020
>
> [14] Deciphering interaction fingerprints from protein molecular surfaces using geometric deep learning, Gainza et al., 2020
>
> [15] Hierarchical, rotation-equivariant neural networks to select structural models of protein complexes, Eismann et al, 2020

---

### Author Response · Authors · 2021-11-15
**General updates and comments**

We thank all reviewers for their extensive feedback and time invested in suggesting important improvements for our work. We also changed the model name to **EquiDock**. We keep the previous name, IEGMN, just for the message passing neural network part of our model.


Our main paper has **significant changes**. Here are our main highlights:

&nbsp;

## Experimental improvements and updates:

- **Significant improvements in model quality** (Table 1): In our updated results, we achieve ~25% RMSD reduction compared to the previous submitted version. We now often outperform the baselines with the exception of HDock. However, as we discuss in the “Complex Prediction Results” paragraph, the baseline scores might be optimistic. These improved results are due to engineering improvements (detailed in footnote [a]; message below).

&nbsp;

- **Two new baselines**: **ClusPro (PIPER)** (Desta, 2020) and **PatchDock** (Mashiach, 2010). In particular, ClusPro is an established method used by the recent concurrent AlphaFold-Multimers (Evans, 2021). We note that these baselines required us to manually submit each of the 125 test examples on their respective web servers, which is a time-consuming process.

&nbsp;

- **More informative plots**: Please see Figures 4, 5, 9, 10, 11 and Appendices C and E.

&nbsp;

- **How accurate are our surface features?** We provide further empirical evidence for the value of using our fast surface features as a high quality proxy for more expensive surface estimation methods. We compute the correlation between our surface feature in Eq. 16 and the (computationally more expensive) MSMS residue’s depth. We show a concentrated distribution of Spearman ranking correlation coefficients around the value of 0.7, suggesting strong correlation with the MSMS values.  See histogram in Figure 10 and discussion in Appendix C. We provide a further synthetic experiment in Figure 9 .

&nbsp;

- **Regarding "[single complex] run times in the order of minutes or even hours do not matter."**


     - We respectfully share a different perspective on the above quoted statement. Our method, EquiDock, can predict the binding of large libraries containing variants of proteins of interest to a target. Depending on the library size, processing a **single complex candidate in seconds vs minutes/hours** is a crucial difference, see also related work on that [11].
         - Example 1: Intensive screening applications that aim to scan over vast search spaces at an unprecedented scale, e.g., for drug discovery.
         - Example 2 (bioengineering): De novo design of proteins binding to specific targets using deep learning methods (e.g., for antibodies [6]).
         - Example 3: Use cases for which protein docking models are just a component of significantly larger end-to-end architectures targeting more involved biological scenarios, for example representing a drug's mechanism of action or modeling cellular processes with a single machine learning model as opposed to a multi-pipeline architecture.

     - We note that all baselines are 80-500 times slower than EquiDock, and such runtimes hinder their integration with other deep learning models, e.g., for predicting sequences of protein-protein interactions. We believe that AI solutions need to be extremely fast in order to scale to the increasing amounts of available experimental data and benefit from the specialized deep learning hardware. Our model takes an important step towards this goal.
     - Please see an updated discussion in the paragraph “Computational Efficiency” at the end of Sec. 5.

&nbsp;

- **Ablations various model components:** Added to our updated paper in Table 5.

&nbsp;
&nbsp;

## EquiDock generality:

- Our model is general and can be applied to other 3D interaction or docking tasks. One example is predicting drug-protein binding from scratch, i.e., jointly predicting the binding pocket and the ligand pose and orientation, without prior knowledge on possible pocket locations. A practical application of this setup is finding targets for new proteins (e.g., mutated viruses), for example to speed up drug repurposing.

&nbsp;

- We have already adapted our EquiDock for the rigid version of this new task, and we outperform QuickVina-W [1] and GNINA [2] on relevant metrics on the PDBbind-CN dataset. However, we believe that the specific details of this new task and the practical importance of modeling ligand flexibility make it more appropriate to discuss these new results and challenges in a separate study.


&nbsp;


Bibliography: message below

---

> ### Author Response · Authors · 2021-11-15
> **General updates and comments (2)**
>
> ## EquiDock wide applicability:
>
> - Our method can be integrated end-to-end to boost the quality of other models (see above discussion on runtime importance). Examples are predicting functions of protein complexes [3] or their binding affinity [5], de novo generation of proteins binding to specific targets (e.g., antibodies [6]), modeling back-bone and side-chain flexibility [4], or devising methods for non-binary multimers. See the updated discussion in the “Conclusion” section of our paper.
>
> &nbsp;
>
> ## Advantages over previous methods:
>
> - Our method does not rely on templates or heavy candidate sampling [7], aiming at the ambitious goal of predicting the complex pose directly. This should be interpreted in terms of generalization (to unseen structures) and scalability capabilities of docking models, as well as their applicability to various other tasks (discussed above).
>
> &nbsp;
>
> - Our method obtains a competitive quality without explicitly using previous geometric (e.g., 3D Zernike descriptors [8]) or chemical (e.g., hydrophilic information) features [3]. Future EquiDock extensions would find creative ways to leverage these different signals and, thus, obtain more improvements.
>
> &nbsp;
> &nbsp;
>
> ## Novelty of theory:
>
> - Our work is the **first** to formalize the notion of **pairwise independent SE(3)-equivariance**. Previous work (e.g., [9,10]) has incorporated only single object Euclidean-equivariances into deep learning models. For tasks such as docking and binding of biological objects, it is crucial that models understand the concept of multi-independent Euclidean equivariances.
>
> - All propositions in Section 3 are **our novel theoretical contributions**.
>
> - We have rewritten the Contribution and Related Work sections to clarify this aspect.
>
> &nbsp;
> &nbsp;
> ---
> Footnote [a]: We have fixed an important bug in the cross-attention code. We have done a more extensive hyperparameter search and understood that layer normalization is crucial in layers used in Eqs. 5 and 9, but not on the $h$ embeddings as it was originally shown in Eq. 10.  We have seen benefits from training our models with a longer patience in the early stopping criteria (30 epochs for DIPS and 150 epochs for DB5). Increasing the learning rate to 2e-4 is important to speed-up training. Using an intersection loss weight of 10 leads to improved results compared to the default of 1.
>
> &nbsp;
>
>
> Bibliography:
>
> [1] Protein-ligand blind docking using QuickVina-W with inter-process spatio-temporal integration, Hassan et al., 2017
>
> [2] GNINA 1.0: molecular docking with deep learning, McNutt et al., 2021
>
> [3] Protein-protein and domain-domain interactions, Kangueane and Nilofer, 2018
>
> [4] Side-chain Packing Using SE(3)-Transformer, Jindal et al., 2022
>
> [5] Contacts-based prediction of binding affinity in protein–protein complexes, Vangone et al., 2015
>
> [6] Iterative refinement graph neural network for antibody sequence-structure co-design, Jin et al., 2021
>
> [7] Hierarchical, rotation-equivariant neural networks to select structural models of protein complexes, Eismann et al, 2020
>
> [8] Protein-protein docking using region-based 3D Zernike descriptors, Venkatraman et al., 2009
>
> [9] SE(3)-transformers: 3D roto-translation equivariant attention networks, Fuchs et al, 2020
>
> [10] E(n) equivariant graph neural networks, Satorras et al., 2021
>
> [11] Fast end-to-end learning on protein surfaces, Sverrisson et al., 2020

---

### Decision · Program_Chairs · 2022-01-20

**Decision:**

Accept (Spotlight)

**Comment:**

This paper introduces a novel SE(3) equivariant graph matching network, along with a keypoint discovery and alignment approach, for the problem of protein-protein docking, with a novel loss based on optimal transport. The overall consensus is that this is an impactful solution to an important problem, whereby competitive results are achieved without the need for templates, refinement, and are achieved with substantially faster run times.